# Secretagogin expression delineates functionally-specialized populations of striatal parvalbumin-containing interneurons

Farid N Garas[1], Rahul S Shah[1], Eszter Kormann[1], Natalie M Doig[1], Federica Vinciati[1], Kouichi C Nakamura[1], Matthijs C Dorst[2], Yoland Smith[3,4], Peter J Magill[1*†], Andrew Sharott[1*†]

[1]Medical Research Council Brain Network Dynamics Unit, Department of Pharmacology, University of Oxford, Oxford, United Kingdom; [2]Department of Neuroscience, Karolinska Institutet, Stockholm, Sweden; [3]Yerkes National Primate Research Center, Department of Neurology, Emory University, Atlanta, United States; [4]Udall Center of Excellence for Parkinson's Disease Research, Emory University, Atlanta, United States

*For correspondence: peter.
magill@pharm.ox.ac.uk (PJM);
andrew.sharott@pharm.ox.ac.uk
(AS)

†These authors contributed
equally to this work

Competing interests: The
authors declare that no
competing interests exist.

Reviewing editor: Rui M Costa,
Fundação Champalimaud,
Portugal

**Abstract** Corticostriatal afferents can engage parvalbumin-expressing (PV+) interneurons to rapidly curtail the activity of striatal projection neurons (SPNs), thus shaping striatal output. Schemes of basal ganglia circuit dynamics generally consider striatal PV+ interneurons to be homogenous, despite considerable heterogeneity in both form and function. We demonstrate that the selective co-expression of another calcium-binding protein, secretagogin (Scgn), separates PV+ interneurons in rat and primate striatum into two topographically-, physiologically- and structurally-distinct cell populations. In rats, these two interneuron populations differed in their firing rates, patterns and relationships with cortical oscillations in vivo. Moreover, the axons of identified PV+/Scgn+ interneurons preferentially targeted the somata of SPNs of the so-called 'direct pathway', whereas PV+/Scgn- interneurons preferentially targeted 'indirect pathway' SPNs. These two populations of interneurons could therefore provide a substrate through which either of the striatal output pathways can be rapidly and selectively inhibited to subsequently mediate the expression of behavioral routines.

## Introduction

Interactions within and between populations of interneurons and spiny projection neurons (SPN) of striatum are critical for the expression of many basal ganglia-dependent behaviors. One major type of GABAergic striatal interneuron expresses parvalbumin (PV). Striatal PV+ interneurons can form axo-somatic synapses with SPNs (*Koos and Tepper, 1999*; *Kubota and Kawaguchi, 2000*), allowing them to powerfully inhibit SPNs of both the so-called 'direct pathway' and 'indirect pathway' (*Gittis et al., 2010*; *Planert et al., 2010*). These interneurons exhibit short-latency responses to powerful excitatory inputs from afferents originating in distinct cortical areas (*Ramanathan et al., 2002*; *Sharott et al., 2012*), providing a mechanism for rapidly stopping or delaying SPN spiking (*Mallet et al., 2005*) and, in turn, modulating striatal outputs. Consistent with this, physiologically-classified (putative) PV+ interneurons in striatum, often called 'fast-spiking interneurons' (FSIs), appear to have a specific role in 'selecting' motor programs, firing most intensely at the choice execution/decision making point of a given task (*Friedman et al., 2015*; *Gage et al., 2010*). The

functional importance of striatal PV+ interneurons is further illustrated by the severe movement deficits that result from their loss or selective disruption (*Burguière et al., 2013*; *Gittis et al., 2011*).

In cortical circuits, PV+ GABAergic interneurons comprise multiple cell types with distinct structural, neurochemical and physiological features (*Klausberger and Somogyi, 2008*). In contrast, striatal PV+ interneurons are usually treated as a single functionally-homogenous population that is enriched within dorsolateral striatum (*Gerfen et al., 1985*; *Kita et al., 1990*). There is some evidence, however, of anatomical, physiological and molecular heterogeneity within this population, suggesting that distinct subpopulations of PV+ interneuron could exist within the striatum. Striatal PV+ interneurons in rats can be broadly divided into two subtypes based on the sizes of their axonal and dendritic fields (*Kawaguchi, 1993*), and the ratio of synapses formed with SPN somata and dendrites varies considerably between individual PV+ cells (*Kubota and Kawaguchi, 2000*). Physiologically, identified PV+ interneurons in rats display highly variable firing rates and patterns in vivo (*Sharott et al., 2012*). It is therefore reasonable to speculate that such variance or diversity could arise from the blinded sampling of functionally-distinct subpopulations of PV+ interneurons. Moreover, studies in mouse striatum indicate that almost half of all PV+ interneurons express serotonin receptor subunit 3A, again highlighting possible functional disparity (*Munoz-Manchado et al., 2016*).

In cortical circuits, combinations of molecular markers have been crucial for defining neuronal diversity. In certain specialized cases, the combinatorial expression of calcium-binding proteins, together with other markers such as transcription factors and neuropeptides, has been used to disambiguate interneuron types that had been previously defined by structural features (*Viney et al., 2013*). Combinations of similar molecules could thus be used to systematically demarcate different populations of PV+ interneurons in striatum. Secretagogin (Scgn), an EF-hand calcium-binding protein (*Rogstam et al., 2007*; *Alpár et al., 2012*), is a particularly promising candidate marker of striatal interneuron heterogeneity, as it is expressed by some (sparsely distributed) non-cholinergic striatal neurons (*Mulder et al., 2009*). Using a combination of immunohistochemistry, stereological cell counting and in vivo recordings of identified interneurons, we demonstrate that Scgn is co-expressed in a segregated and specialized population of striatal PV+ interneurons in rats and primates, but not in mice. Our findings suggest that distinct subpopulations of PV+ interneurons could enable corticostriatal afferents to orchestrate SPN activity in a topographically- and output pathway-selective manner.

## Results

### Selective expression of secretagogin divides the PV+ interneuron population in rat striatum but not in mouse striatum

In the adult mouse, the calcium-binding protein secretagogin (Scgn) is expressed by some GABAergic, but not cholinergic, striatal neurons (*Mulder et al., 2009*). To test whether these neurons also express PV, we examined the expression of Scgn and PV across the mouse dorsal striatum using unbiased stereological methods. All striatal Scgn+ cells co-expressed the 'pan-neuronal' marker NeuN, but not the SPN-specific marker Ctip2 (*Arlotta et al., 2008*) (*Figure 1A,B*), indicating that Scgn is exclusively expressed by interneurons. However, interneurons that co-expressed PV and Scgn were rare. Indeed, the vast majority of Scgn+ interneurons did not co-express PV and vice versa (*Figure 1C–F*, *Figure 1—source data 1*). Thus, the previously identified population of GABAergic Scgn+ neurons in mice (*Mulder et al., 2009*) are interneurons, but not PV-expressing interneurons.

Secretagogin is expressed in many brain areas, and expression patterns may vary across species (*Alpár et al., 2012*). We next tested whether the patterns of Scgn expression in mice also held true for rats. As in mice, Scgn-expressing cells in rat dorsal striatum were interneurons (*Figure 1G,H*). However, in comparison to mice, Scgn+ interneurons in rat dorsal striatum could more readily be observed, with or without co-expression of PV (*Figure 1I,J*, *Figure 1—source data 1*). Across all striatal sections, about one half of Scgn+ interneurons also expressed PV (*Figure 1K*, *Figure 1—source data 1*), and about one quarter of PV+ interneurons also expressed Scgn (*Figure 1L*, *Figure 1—source data 1*). These findings suggest that, in rats, Scgn is a candidate marker for a subtype of PV-expressing interneuron. Moreover, Scgn+ interneurons were relatively common overall, with a

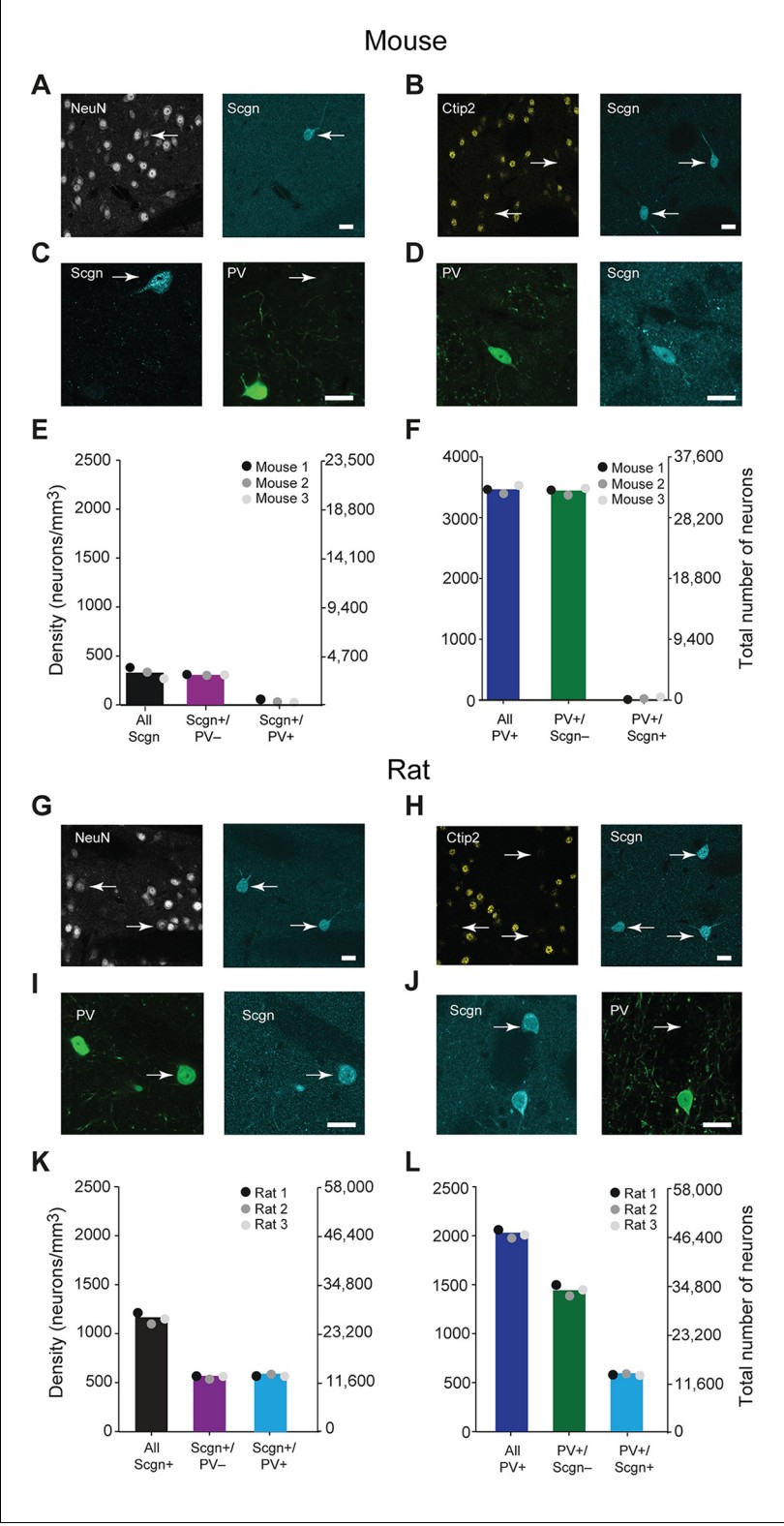

**Figure 1.** A large subpopulation of PV+ interneurons co-express Scgn in the dorsal striatum of rats, but not in mice. (A–F), Co-expression of PV and Scgn in the dorsal striatum of the mouse. (A, B) Immunofluorescence signals for Scgn (arrows), the pan-neuronal marker NeuN (A) and SPN-specific marker, Ctip2 (B). Scgn+ cells express NeuN, but not Ctip2, indicating they are interneurons. (C, D), Immunofluorescence signals for PV and Scgn. The vast majority of Scgn+ neurons did not co-express PV (C, white arrow) although a few did (D). (E, F) Densities and

*Figure 1 continued on next page*

*Figure 1 continued*

numbers of interneurons expressing combinations of PV and/or Scgn. PV and Scgn were seldom expressed by the same interneurons in the mouse dorsal striatum. (**G–L**) As in **A–F**, but all data are from the dorsal striatum of rat. (**G, H**) Scgn+ cells express NeuN, but not Ctip2, indicating they are interneurons. (**I, J**) In contrast to mouse, Scgn+ interneurons that co-expressed PV were common (**I**, white arrow), as were those that did not (**J**, white arrow,). (**K, L**) Densities and numbers of interneurons expressing combinations of PV and/or Scgn in the rat. About one half of Scgn+ interneurons co-expressed PV (**K**). About one quarter of PV+ interneurons co-expressed Scgn (**L**). (Scale bars **A–D** and **G–J**, 20 µm)

The following source data is available for figure 1:

**Source data 1.** Source data for *Figures 1E,F,K,L*.

density that was about two thirds of that of PV+ interneurons, indicating that Scgn is itself a novel marker of a major class of striatal interneuron in the rat. We found this divergence in Scgn immuno-reactivity between rodent species using two antibodies raised against different epitopes of Scgn that have 100% and 95% sequence homologies for rat and mouse, suggesting that the relative paucity of Scgn+ interneurons in mice was not a result of differences in antibody specificity (*Table 1*).

Striatal afferents (and efferents) are topographically organized (*Mailly et al., 2013*; *McGeorge and Faull, 1989*). Thus, if a given cell population has a biased spatial distribution across striatum, it will likely receive privileged inputs from a specific subset of all striatal afferents. With this in mind, we next tested whether the two molecularly-distinct populations of PV+ interneuron, i.e. those that co-expressed Scgn (PV+/Scgn+) and those that did not (PV+/Scgn-), were preferentially localized to discrete striatal regions in the rat. The density of PV+/Scgn- interneurons remained relatively constant along the rostro-caudal axis of dorsal striatum, at least until the most caudal aspects where density decreased by around 75% (*Figure 2Ai,B*, *Figure 2—source data 2*). However, PV+/Scgn+ interneurons displayed a strikingly different pattern of localization, being more sparsely

**Table 1.** Primary antibodies used in this study.

| Molecular Marker | Host Organism | Dilution used | Source and catalog number | Research Resource Identifier (RRID) |
|---|---|---|---|---|
| Choline acetyltransferase | Goat | 1:500 | Millipore AB144P | RRID:AB_2079751 |
| Ctip2 | Rat | 1:500 | Abcam AB18465 | RRID:AB_2064130 |
| Dopamine and cAMP-regulated phosphoprotein-32 (DARPP-32) | Goat | 1:100 | Santa Cruz Biotechnology SC-8483 | RRID:AB_639002 |
| Gephyrin | Mouse | 1:500 | Synaptic Systems 147021 | RRID:AB_2232546 |
| Neuron-specific nuclear antigen (NeuN) | Mouse | 1:200 | Millipore MAB377 | RRID:AB_2298772 |
| Nitric Oxide Synthase | Mouse | 1:500 | Sigma N2280 | RRID:AB_260754 |
| Neuropeptide Y | Rabbit | 1:5000 | ImmunoStar 22940 | RRID:AB_2307354 |
| Parvalbumin | Guinea Pig | 1:1000 | Synaptic Systems 195004 | RRID:AB_2156476 |
| Preproenkephalin | Rabbit | 1:5000 | LifeSpan Biosciences LS-C23084 | RRID:AB_902714 |
| Secretagogin | Goat | 1:1000 | R&D Systems AF4878 | RRID:AB_2269934 |
| Secretagogin | Rabbit | 1:500 | Abcam AB111871 | RRID:AB_10864618 |

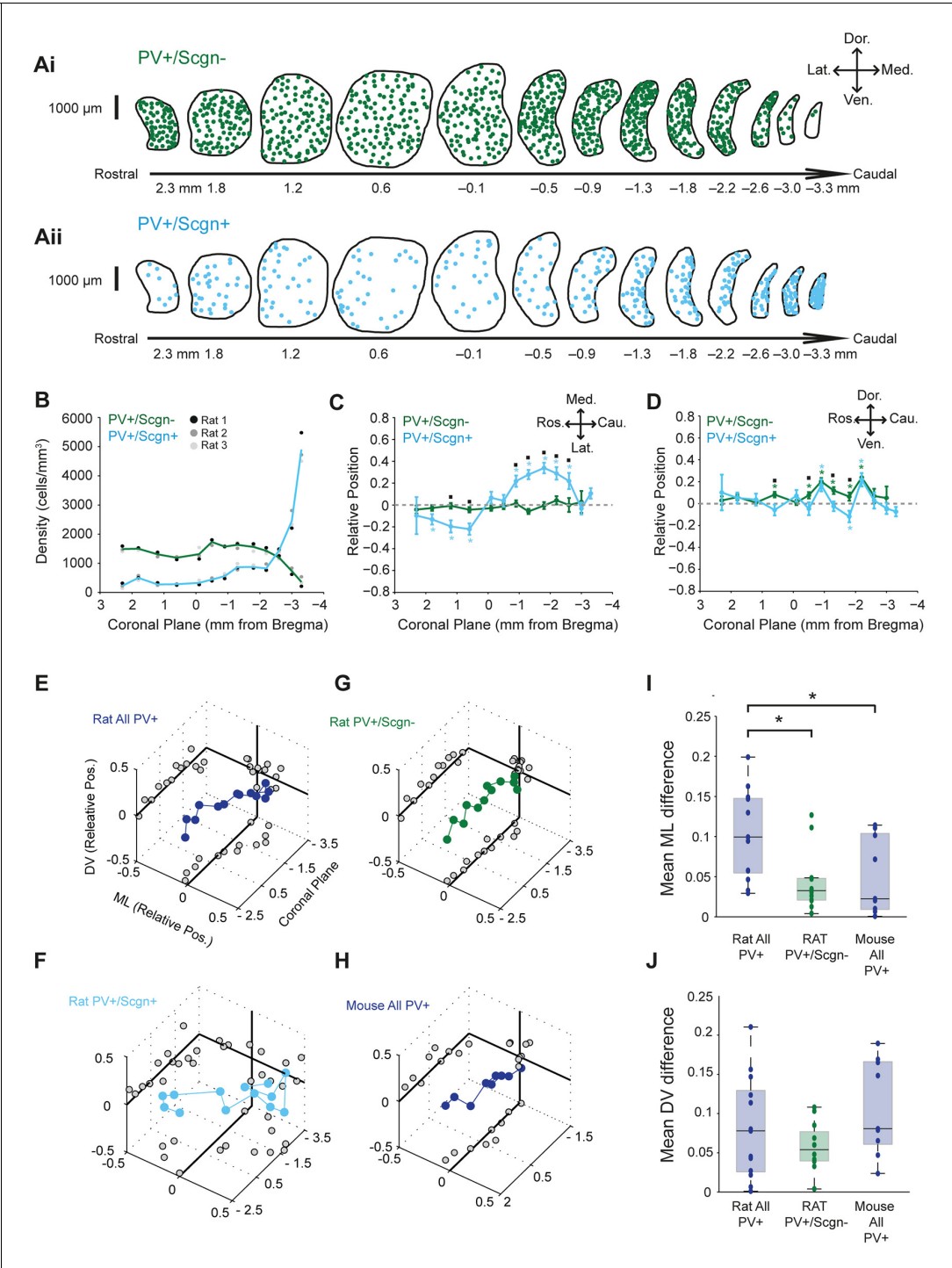

**Figure 2.** PV+/Scgn+ and PV+/Scgn- interneurons have different biases in their distributions in the dorsal striatum of the rat. (A) Typical distributions of PV+/Scgn- interneurons (Ai) and PV+/Scgn+ interneurons (Aii) across 13 coronal planes encompassing the dorsal striatum in rat, with each dot representing a single neuron. (B) Densities of PV+/Scgn+ and PV+/Scgn- interneurons along the rostro-caudal axis of striatum. Note the increase in density of PV+/Scgn+ interneurons, and the small decrease in density of PV+/Scgn- interneurons, from rostral to caudal striatum. C,D Medio-lateral (C) and dorso-ventral (D) distributions of PV+/Scgn- interneurons (green line) and PV+/Scgn+ interneurons (blue line) along the 13 coronal planes. The presence of the asterisk (*) indicates a distribution that is significantly biased in one direction along the specified axis. Squares (□) indicate a significant difference in the distribution of the two populations of PV+ interneurons along the specified axis within a given coronal plane. Data are means of the normalized positions of allneurons counted ± SEMs. (E–G) Mean normalized medio-lateral (ML) and dorso-ventral (DV) positions of all PV+ (E), PV+/Scgn+ (F) and PV+/Scgn- (G) interneurons in each of 13 coronal planes (with respect to Bregma) of rat dorsal striatum. Grey circles on the end panels

*Figure 2 continued on next page*

*Figure 2 continued*

show the distribution of the mean values in each dimension. Note that clear biases in ML positions of all PV+ interneurons are similar to those of PV+/ Scgn+ interneurons, whereas PV+/Scgn- interneurons are not clearly biased. (H) Mean normalized medio-lateral and dorso-ventral co-ordinates of all PV + interneurons in each of 9 coronal planes of mouse dorsal striatum. Note that their positions are not clearly biased. (I, J) The difference in mean medio-lateral positions (I), but not dorso-ventral positions (J), between coronal planes of all rat PV+ interneurons is significantly higher than that of rat PV+/Scgn- interneurons and all PV+ interneurons in the mouse (Kruskal-Wallis ANOVA on ranks with post-hoc Dunn tests).

The following source data is available for figure 2:

**Source data 1.** Source data for *Figures 2 B–H*.

distributed across most striatal levels, except in the most caudal aspects where their density was around three times higher than that of PV+/Scgn- interneurons in any other plane (*Figure 2Aii,B*, *Figure 2—source data 1*). The two populations of interneurons were also differentially distributed along the medio-lateral axis of the striatum; PV+/Scgn- neurons were evenly distributed, whereas PV +/Scgn+ interneurons tended to distribute more laterally in the rostral striatum but more medially in caudal striatum (*Figure 2C*, *Supplementary file 1*, *Figure 2—source data 1*). This particular pattern of bias in the distribution of PV+/Scgn+ interneurons is highly unusual in that it has not been described for any other striatal cell population. Along the dorso-ventral axis, PV+/Scgn- interneurons were more likely than PV+/Scgn+ interneurons to display a slight dorsal bias in their localization (*Figure 2D*, *Supplementary file 1*, *Figure 2—source data 1*). Taken together, these data show that the selective expression of Scgn distinguishes two populations of PV+ interneurons that display significantly different spatial distributions in the dorsal striatum of the rat.

Given that PV+/Scgn+ and PV+/Scgn- interneurons are not equally abundant (*Figure 1F,L*), we next estimated whether their different spatial distributions biased the distribution of all PV+ interneurons as a whole. When all rat PV+ interneurons were grouped together for analysis, and all coronal planes were considered, there was no consistent bias in their relative positions along the dorso-ventral axis (*Figure 2E*, *Figure 2—source data 1*). However, the relative medio-lateral positions of all PV+ interneurons varied along the rostral-caudal axis; they tended to be laterally positioned in those coronal planes rostral of Bregma, but medially positioned in planes caudal to Bregma (*Figure 2E*). The distribution biases of all PV+ interneurons were thus similar to those of PV+/Scgn+ interneurons (*Figure 2F*, *Figure 2—source data 1*), suggesting the presence of PV+/Scgn+ interneurons biases the entire PV+ interneuron population. In line with this, there was no consistent or strong bias in the relative positions of PV+/Scgn- interneurons (*Figure 2G*, *Figure 2—source data 1*). Taken together, these data suggest that, in rat dorsal striatum, the biased medio-lateral distributions of PV+ interneurons can be largely explained by the selective expression of Scgn. To further explore this notion, we analyzed the spatial distributions of all PV+ interneurons in the mouse dorsal striatum (*Figure 2H*), which contains a tiny number of PV+/Scgn+ interneurons (*Figure 1F*). There was no consistent or strong bias in the relative positions of all PV+ interneurons in mice (*Figure 2H*) and, as such, their distribution pattern closely resembled that of PV+/Scgn- interneurons in rats (*Figure 2G*), but not that of all PV+ interneurons in rats (*Figure 2E*). These observations were supported by quantitative comparisons of the differences in medio-lateral positions across coronal planes of all mouse PV+ interneurons, all rat PV+/Scgn- interneurons and all rat PV+ interneurons; the differences in the latter were significantly larger (*Figure 2I*). The relatively uniform distribution of PV+ interneurons in mouse striatum thus tallies with their relative lack of Scgn expression. These data serve to reinforce that Scgn is a useful and highly-relevant marker for dividing PV+ interneuron populations.

In summary, rat PV+/Scgn+ interneurons have a highly unusual spatial distribution, transitioning from predominantly lateral to medial positions as the rostro-caudal axis of dorsal striatum is traversed; their biased positioning accounts for much of the non-uniform distribution of all PV+ interneurons in rats. However, in contrast to a previous report (*Luk and Sadikot, 2001*), we found no quantitative evidence of a consistent 'dorsolateral' bias in the distribution of all PV+ interneurons in the striatum of rats. This discrepancy presumably arises from differences in the cell-counting methodologies and analyses used, including the extent to which different regions of dorsal striatum were sampled; we calculated the relative position of each neuron counted in 13 coronal planes, rather

than calculating cell densities according to arbitrary 'quadrants' in 1 coronal plane (Luk and Sadikot, 2001).

## Selective secretagogin expression divides the PV+ interneuron population in the primate caudate and putamen

Our data in rats and mice show that the co-expression of Scgn by striatal PV+ interneurons is not highly conserved across rodent species. Because Scgn+ neurons have also been reported in the primate striatum (*Mulder et al., 2009*), we further explored the possibility of phylogenetic conservation by analyzing the co-expression of PV and Scgn in interneurons of the monkey (rhesus macaque) striatum. Secretagogin was expressed by some neurons in both the caudate nucleus and putamen of monkeys; it was frequently co-expressed with PV (*Figure 3A–C*, *Figure 3—source data 1*). Compared to the rat, co-expression of PV and Scgn was more common in primates; about three quarters of PV+ interneurons in caudate nucleus and putamen also expressed Scgn (*Figure 3C*, *Figure 3— source data 1*). Thus, Scgn is itself a novel marker of a major class of striatal interneuron in both monkeys and rats. Along the rostro-caudal axis of monkey striatum, but most notably in the caudate nucleus, there was a marked increase in the density of PV+/Scgn+ interneurons in caudal planes (*Figure 3D*, *Supplementary file 2*, *Figure 3—source data 1*), mirroring the biased rostro-caudal distribution of PV+/Scgn+ interneurons in the rat striatum (*Figure 2B*). The density of PV+/Scgn- interneurons remained relatively constant across the rostro-caudal extent of monkey striatum, which is again in line with our observations in rats (*Figure 3D*, *Supplementary file 2*, *Figure 3—source data 1*.). Interestingly, the distribution of PV+/Scgn- interneurons, but not PV+/Scgn+ neurons, was laterally biased throughout rostro-caudal aspect putamen, but not caudate (*Figure 3E*, *Supplementary file 2*, *Figure 3—source data 1*). When taken together, the data from monkey and rat not only show that a substantial proportion of striatal PV+ interneurons co-express Scgn, a population enriched in caudal striatum, but also that these novel constituents of the striatal microcircuit are phylogenetically conserved to some extent.

## Spontaneous in vivo activity of identified PV+/Scgn- and PV+/Scgn+ interneurons during cortical SWA and activation states

As previously described, PV+ interneurons display the most diverse in vivo firing rates/patterns of the major striatal interneuron types in the rat (*Sharott et al., 2012*). We next investigated whether the selective expression of Scgn could account for any of the variability in the activity of striatal PV+ interneurons. We thus extracellularly recorded the action potentials fired by individual interneurons in the dorsal striatum of anesthetized rats, and then juxtacellularly labeled the same interneurons with neurobiotin for *posthoc* verification of their neurochemical identities and locations (*Figure 4*). We focused our analyses on the firing of identified PV+ interneurons during two distinct brain states, slow-wave activity (SWA) and 'cortical activation' (*Sharott et al., 2012*), as defined by simultaneous recordings of the electrocorticogram (ECoG).

Striatal PV+ interneurons have short duration (<1 ms) action potentials (*Mallet et al., 2005*; *Sharott et al., 2012*), a characteristic often used to putatively identify them (as FSIs) in awake, behaving animals (*Adler et al., 2013*; *Berke, 2004*). In agreement with this, our recordings confirmed that the action potential waveforms of all PV+ interneurons, irrespective of Scgn immunoreactivity, were brief (1.09 ± 0.06 ms, n = 24; *Figure 5A*.). Further analysis showed that the duration of the first deflection (D1) of the action potentials (see *Figure 5A*) of PV+/Scgn+ interneurons (n = 10 cells; 0.31 ± 0.014 ms) was significantly shorter than those of PV+/Scgn- interneurons (n = 14 cells; 0.39 ± 0.02 ms) (Mann Whitney, p=0.006). Although the firing rates of PV+ interneurons as a whole could vary substantially (range: 0.01 – 22.52 spikes/s; see *Figure 4*) during SWA, the average firing rates of PV+/Scgn+ and PV+/Scgn- interneurons were similar in this brain state (*Figure 5B*, *Figure 5—source data 1*). However, during cortical activation, the firing rate of PV+/Scgn+interneurons (median = 13.8 spikes/s) was significantly higher than that of the PV+/Scgn- interneurons (median = 3.74 spikes/s, Mann-Whitney U test, p=0.03; *Figure 5B*, *Figure 5—source data 1*).

Striatal PV+ interneurons display considerable heterogeneity in their firing patterns, more so than other striatal interneuron types (*Sharott et al., 2012*). In order to quantify this observation, we calculated correlations between the interspike interval (ISI) distributions of pairs of interneurons of the same type (*Figure 5C–H*). We analyzed firing pattern homogeneity of all PV+ interneurons

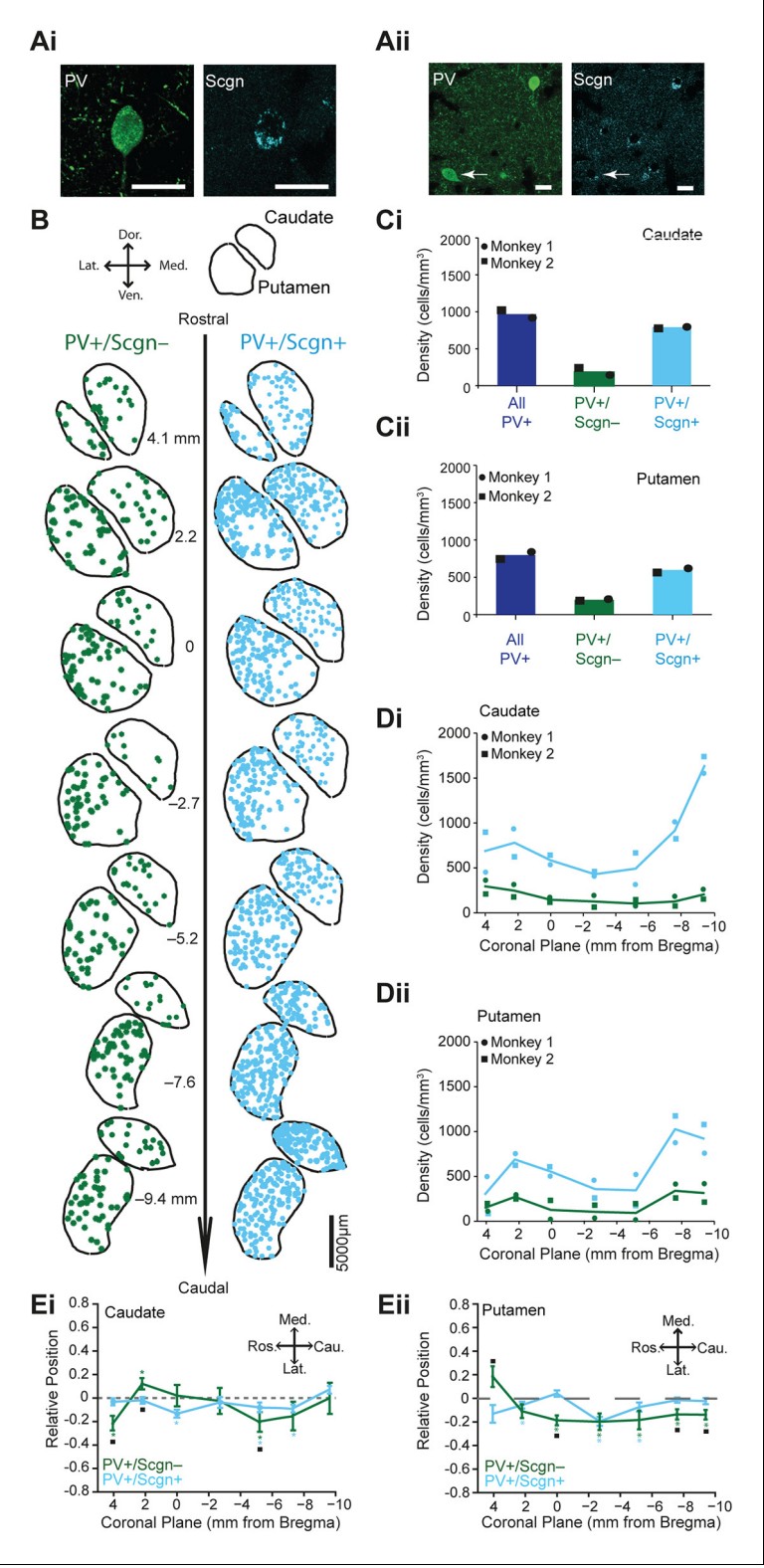

**Figure 3.** PV+/Scgn+ and PV+/Scgn- interneurons have different biases in their distributions in the caudate and putamen of macaque monkey. (**Ai, Aii**) Confocal micrographs of the macaque striatum showing PV-expressing interneurons that co-expressed (**Ai**) and did not co-express (**Aii**) Scgn (arrow). (**B**) Typical distributions of PV+/Scgn-interneurons (left) and PV+/Scgn+ interneurons (right) across 7 coronal planes of macaque caudate and putamen, with each dot representing a single neuron. (**C**) Mean densities of all PV+ interneurons across the entirety of the

*Figure 3 continued on next page*

*Figure 3 continued*

caudate nucleus (**Ci**) and the putamen (**Cii**), including those populations that co-express Scgn (blue) and do not express Scgn (green). In the macaque caudate-putamen, PV+/Scgn+ neurons represent nearly three quarters of all PV-expressing neurons. Dots and squares indicate the values for individual animals. (**D**) Densities of PV+/Scgn+ interneurons (blue) and PV+/Scgn- interneurons (green) along the rostro-caudal axis of the caudate (**Di**) and the putamen (**Dii**). Note that the PV+/Scgn+ population in the macaque increases in density towards the caudal planes of the caudate and the putamen. (**E**) Medio-lateral distribution of PV+/Scgn+ and PV+/Scgn- interneurons along 7 coronal planes of the caudate nucleus (**Ei**) and the putamen (**Eii**). The presence of the asterisk (*) indicates a distribution that is significantly biased in one direction along the specified axis. Squares (□) indicate a significant difference in the distribution of the two PV+ interneuron populations along the specified axis within a given coronal plane. Data are means of the position of all neurons counted ± SEMs.

The following source data is available for figure 3:

**Source data 1.** Source data for *Figures 3B–E*.

---

(irrespective of Scgn expression) as compared to that of cholinergic (ChAT+) interneurons, which fire with a relatively narrow range of ISIs (*Sharott et al., 2012*). In line with our previous observations, correlations between the ISI histograms of ChAT+ interneurons often result in r-values close to 1 (*Figure 5C,E*). In contrast, while correlations between PV+ interneurons could also yield strong correlations, many ISI histograms differed, thereby producing lower r-values (*Figure 5D,F*). Across all recorded neurons, the ISI histograms of ChAT+ interneurons were significantly more positively correlated than those of PV+ interneurons during both SWA and cortical activation (Mann Whitney, SWA; $p=0.001 \times 10^{-37;}$ Act: $p<0.025 \times 10^{-36}$), confirming that PV+ interneurons have more heterogeneous firing patterns than ChAT+ interneurons.

If PV+ interneurons encompass multiple neuron types, and Scgn is a marker of a single subpopulation of PV+ interneurons, pairs of PV+/Scgn+ interneurons should display more homogeneity in their firing patterns than the population as a whole. To test this hypothesis, we computed correlations between the ISI histograms of pairs of PV+/Scgn+ interneurons and pairs of PV+/Scgn- interneurons. During both SWA and cortical activation (*Figure 5G,H*, *Figure 5—source data 1*), the r-values for pairs of PV+/Scgn+ interneurons were significantly higher than those of PV+/Scgn- pairs ($p<0.05$ for both Kruskal–Wallis ANOVAs and *post hoc* Dunn's tests), but not significantly different from pairs of ChAT+ interneurons. These results suggest that, across brain states, the homogeneity of firing patterns within the subpopulation of PV+/Scgn+ interneurons was akin to that of cholinergic interneurons and greater than within the subpopulation of PV+/Scgn- interneurons.

## Firing of PV+/Scgn- and PV+/Scgn+ interneurons is preferentially phase-locked to different cortical network oscillations

Identified striatal PV+ interneurons, as well as FSIs, show a strong tendency to phase lock their firing to cortical oscillations (*Berke, 2004*, *2009*; *Sharott et al., 2012*). Thus, we next examined whether PV+/Scgn- and PV+/Scgn+ interneurons fired differently with respect to the phase of cortical population oscillations (as recorded in ipsilateral, frontal ECoG) across frequencies from 0.4–80 Hz in SWA and cortical activation (*Figure 6*). As suggested by the raw data (*Figure 4*), both PV+/Scgn- and PV+/Scgn+ interneurons tended to fire around the peaks of cortical slow oscillations (0.4–1.6 Hz) during SWA (*Figure 6A,B*). Although the firing of all PV+ interneurons was significantly locked to cortical slow oscillations to some extent (*Figure 6C*, *Figure 6—source data 1*), the locking across the population was stronger in the PV+/Scgn- neurons (*Figure 6A,B*). In line with these results, the vector length of firing of PV+/Scgn- interneurons was around twice that of PV+/Scgn+ interneurons (*Figure 6D*; Mann Whitney, $p=0.04$). Similarly, the firing of PV+/Scgn- interneurons was more strongly locked to cortical spindle oscillations (7–12 Hz), which was reflected in both a greater number of significantly locked neurons (*Figure 6C*, *Figure 6—source data 1*) and greater vector length (*Figure 6D*, Mann Whitney, $p=0.008$). In contrast, the firing of PV+/Scgn+ interneurons was more tightly locked to cortical gamma (30–80 Hz) oscillations (*Figure 6A,B*), and a greater proportion of PV+/Scgn+ interneurons were significantly locked to gamma oscillations (*Figure 6C*). The phase-locked firing of PV+/Scgn- and PV+/Scgn+ interneurons was generally more similar across all

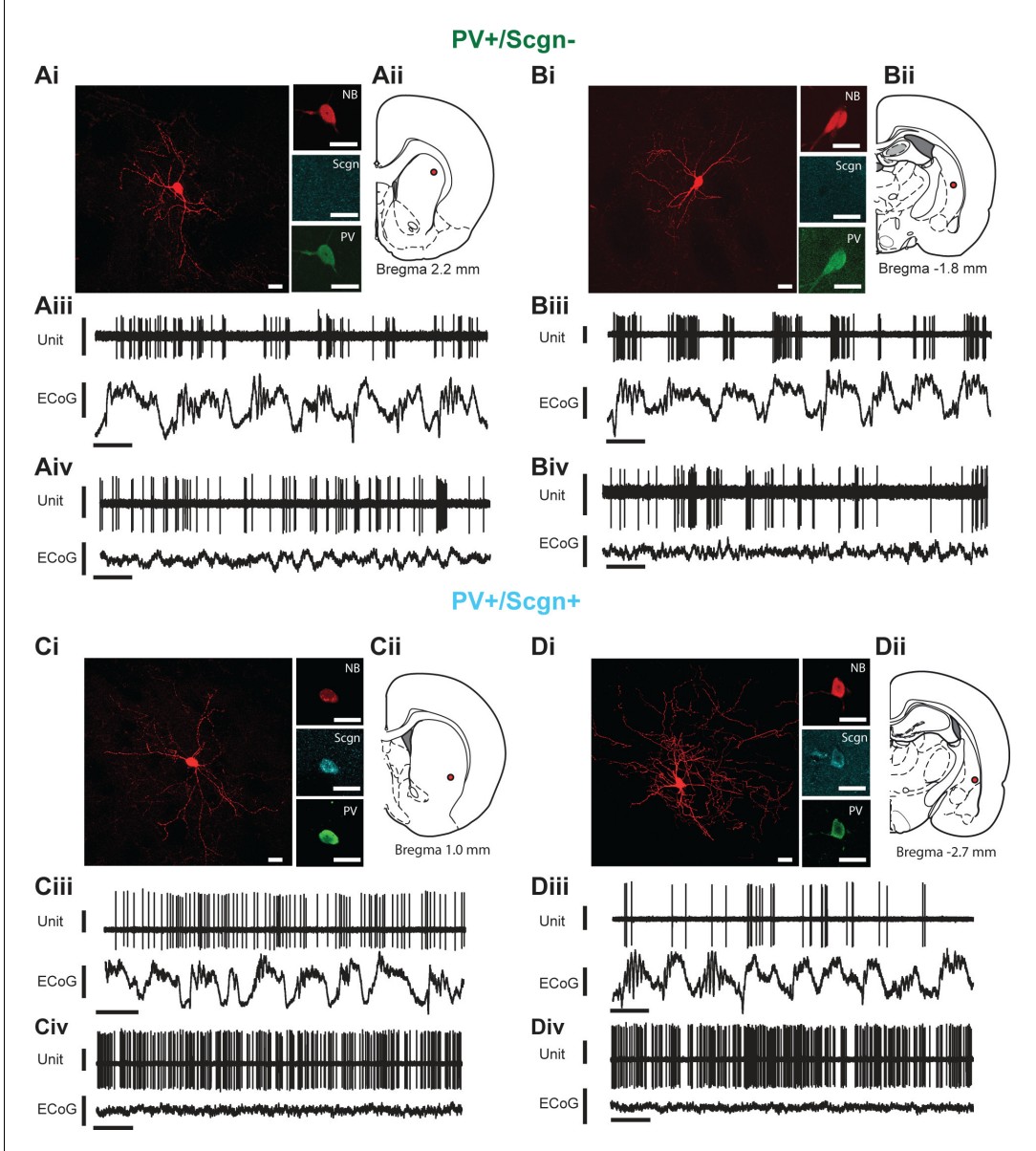

**Figure 4.** In vivo electrophysiological properties of identified PV+/Scgn- and PV+/Scgn+ striatal interneurons in the rat. (**A,D**) Juxtacellularly-labeled PV +/Scgn- interneurons (**A,B**) and PV+/Scgn+ interneurons (**C,D**), identified by their co-localization of fluorescent labeling for neurobiotin (NB) and calcium-binding proteins. PV+ interneurons were recorded in the rostral (**Aii**), central (**Cii**) and caudal (**Bii, Dii**) aspects of dorsal striatum. (**Aiii–Diii**) Spontaneous action potential discharges (unit activity) of the same individual PV+/Scgn- and PV+/Scgn+ interneurons during robust cortical slow-wave activity (SWA), defined using the frontal electrocorticogram (ECoG). (**Aiv–Div**) Firing of the same interneurons during spontaneous cortical activation. (**Ai–Di**) Scale bars are 20 µM. (**Aii–iii-Dii-iii**) Vertical scale bars for unit activity are 0.5 mV; Vertical scale bars for ECoG are 1 mV; Horizontal scale bars are 1 s).

cortical oscillation frequencies during the activated brain state (*Figure 6E,F,H*, *Figure 6—source data 1*). However, around three times as many PV+/Scgn+ interneurons were locked at gamma frequencies between 30 and 60 Hz as compared to PV+/Scgn- interneurons (*Figure 6G*, *Figure 6— source data 1*). These results indicate that the temporal organization of the firing of PV+/Scgn- and PV+/Scgn+ interneurons with respect to ongoing cortical oscillations is distinct and brain state-dependent, thus demonstrating further physiological divergence between these cell populations.

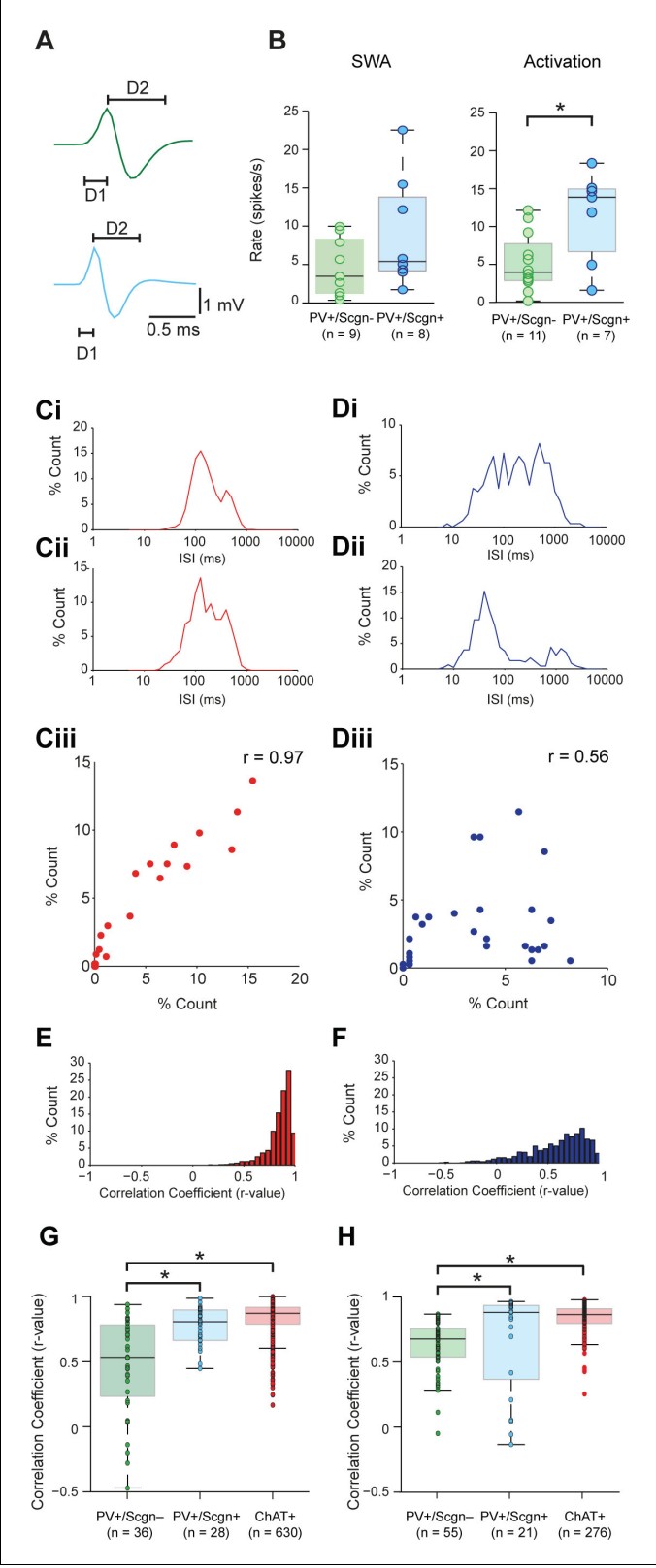

**Figure 5.** Identified PV+/Scgn- and PV+/Scgn+ interneurons in the dorsal striatum of rats have distinct electrophysiological properties. (**A**) The mean action potential waveforms of PV+/Scgn-(green) and PV+/Scgn +(blue) interneurons. Both groups had short waveforms (around 1 ms). Waveforms were split in to D1 (baseline to its peak) and D2 (return to the baseline from peak). The D1 segment of PV+/Scgn+interneurons was significantly

*Figure 5 continued on next page*

*Figure 5 continued*

shorter than that of PV+/Scgn- interneurons. (**B**) The median firing rates of the two populations of PV+ interneuron were similar during SWA, but PV+/Scgn+ interneurons had a significantly higher firing rate than PV+/Scgn- interneurons during cortical activation. (**C–F**) The similarity of firing patterns between pairs of PV+/Scgn-, PV+/Scgn+ and cholinergic (ChAT+) interneurons was assessed by calculating the correlation coefficient (Spearman) between interspike interval (ISI) histograms. (**Ci–ii**) ISI histograms from the spike trains for two cholinergic interneurons recorded separately during cortical activation. (**Ciii**) Scatter plot of the % count in each ISI bin for the two ChAT+ interneurons plotted against each other. Because of the similar (unimodal) ISI distributions, the Spearman Rho correlation coefficient (r) for the pair of ChAT+ interneurons is relatively high. (**Di–ii**) As in C, but the analysis is now performed on two separately recorded PV+ interneurons. (**Diii**) Because of the more variable ISI distributions, the correlation coefficient (r) is relatively low. (**E**) Histogram of the correlation coefficients of 276 cholinergic interneuron pairs from recordings during cortical activation. The majority of correlations are high, indicating similar ISI histograms across the population. (**F**) Histogram of the correlation coefficients of 325 PV+ interneuron pairs from recordings during cortical activation. Correlations strengths are relatively spread; indicating ISI histograms are less similar than those of the cholinergic population. (**G, H**) Comparison of Spearman correlation coefficients of all PV+/Scgn-, PV+/Scgn+ and ChAT+ interneuron pairs recorded in SWA (**G**) and cortical activation (**H**). In both brain states, the correlation coefficients between pairs of PV+/Scgn+ ISI histograms was significantly higher than that of PV+/Scgn- pairs, but not significantly different to ChAT+ interneuron pairs.

The following source data is available for figure 5:

**Source data 1.** Source data for *Figure 5A,B,G,H*.

## PV+/Scgn- and PV+/Scgn+ interneurons differ in their somatic targeting of striatal projection neurons of the direct and indirect pathways

The axon terminals of dorsal striatal PV+ interneurons/FSIs often form multiple "basket-like" appositions with SPN cell bodies (*Koos and Tepper, 1999*; *Kubota and Kawaguchi, 2000*), allowing them to powerfully inhibit SPNs, a mechanism which is thought central to their role in the striatal microcircuit (*Tepper and Bolam, 2004*). Fully delineating the role of any neuron type requires an understanding of not only its neurochemical/molecular properties and the temporal organization of its activity, but also of the cellular targets that it innervates. With the latter in mind, some of the recorded and neurobiotin-labeled PV+/Scgn+ and PV+/Scgn- interneurons (n = 4 and 4, respectively) were used to gain insight into whether these two cell populations target the same types of SPN to the same extent. We thus compared the prevalence of appositions of the terminal-like axonal varicosities ('boutons') of neurobiotin-labeled PV+/Scgn+ or PV+/Scgn- interneurons with the cell bodies of SPNs (selectively labeled with DARPP-32). In agreement with previous descriptions (*Kawaguchi, 1993 Sharott et al., 2012*), PV+ interneurons exhibited dense local axonal arborizations (see *Figure 4*) that often formed varicosities in close proximity to (i.e. were apposed to) the somata of SPNs (*Figure 7*). Whether or not these appositions indicated the presence of functional axo-somatic synapses between PV+ interneurons and SPNs was investigated by simultaneously detecting the presence of gephyrin, which is highly enriched in the post-synaptic membranes of GABAergic synapses (*Sigal et al., 2015*). Quantification of the overlap between gephyrin puncta and axonal varicosities revealed that appositions between neurobiotin-labeled axonal boutons and SPNs were often the sites of putative GABAergic synapses (*Figure 7*). Indeed, 69.2 ± 2.4% of appositions (n = 104) made by the axons of PV+/Scgn+ interneurons (n = 3) with SPN somata were associated with discrete puncta of gephyrin immunoreactivity, while 75.5 ± 6.6% of appositions (n = 94) made by the axons of PV+/Scgn- neurons (n = 2) were associated with discrete gephyrin+ puncta. These data suggest that, at the site of an apposition of a PV+ interneuron axonal bouton with a SPN soma, there is a high probability of a GABAergic synapse being formed.

Previous work has shown that the proportion of somatic (as compared to dendritic) synapses formed by the axons of individual PV+ interneurons ranges from 7 to 58% (*Kubota and Kawaguchi, 2000*). Next we examined whether any of this variability could be explained by systematic differences in innervation of SPN somata by the two populations of PV+ interneuron. Neurobiotin-labeled boutons of PV+/Scgn- interneurons (neurons = 4, boutons = 347) were almost twice as likely as those of PV+/Scgn+ interneurons (neurons = 3, boutons = 709) to appose the somata of SPNs (*Figure 8C*;

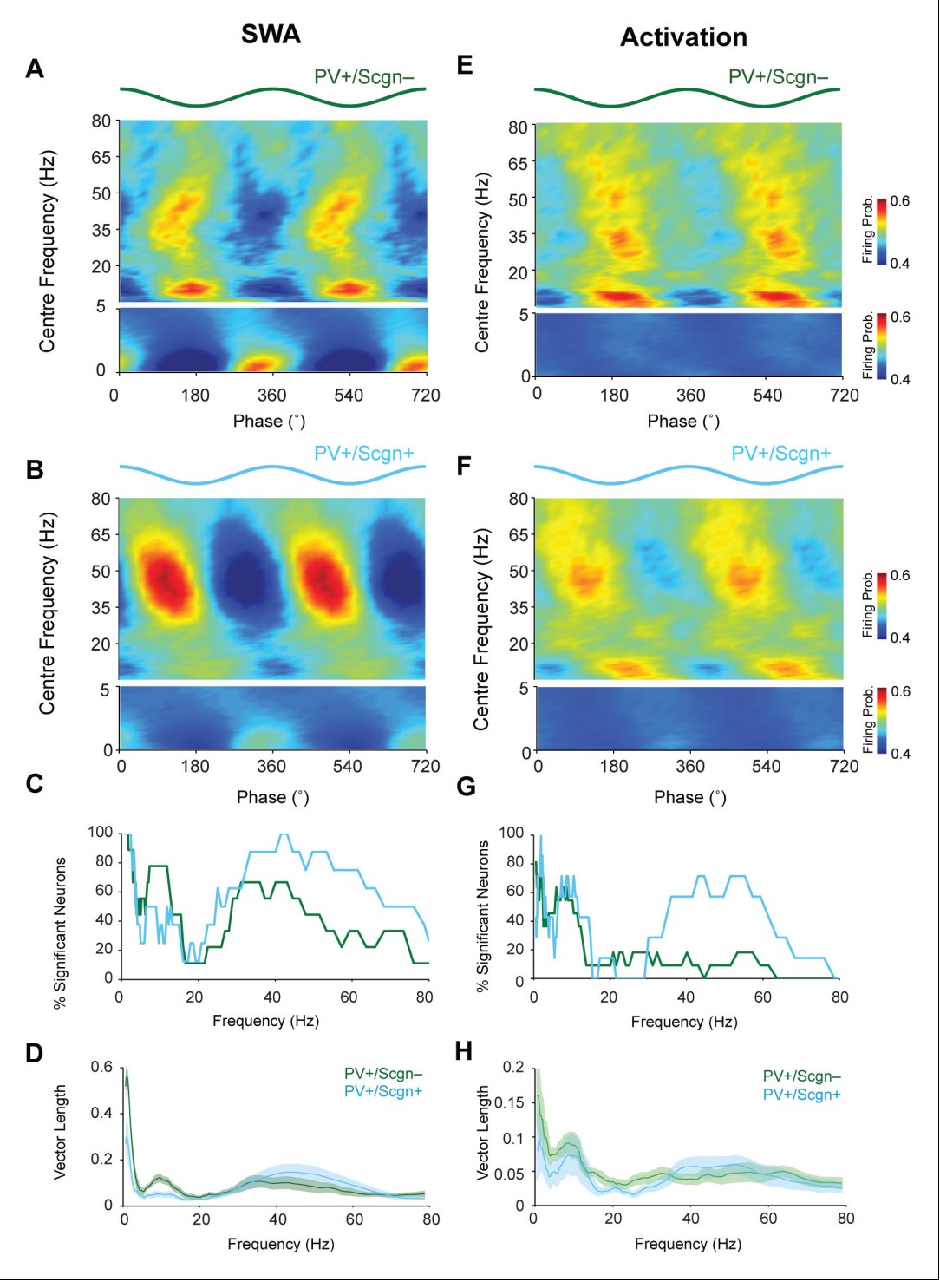

**Figure 6.** The firing of PV+/Scgn- and PV+/Scgn+ striatal interneurons is distinctly phase locked to cortical oscillations in the rat. (A,B) Mean phase histograms of the firing of striatal PV+/Scgn- (A) and PV+/Scgn+ (B) interneurons with respect to cortical oscillations of 0.4–80 Hz during SWA. Note the stronger locking of the firing of PV+/Scgn- interneurons to slow (0.4–1.6 Hz) and spindle (7–12 Hz) frequencies, and the stronger locking of PV+/Scgn+ interneurons to gamma oscillations (30–80 Hz) (C) Histogram showing the proportions of PV+/Scgn- (green) and PV+/Scgn+ (blue) interneurons that exhibited significantly phase-locked firing (as measured by the Raleigh test, with p<0.05) in each frequency range of cortical oscillation during SWA. (D) Mean vector lengths calculated across all PV+/Scgn- (green) and PV+/Scgn+ (blue) neurons recorded during SWA (PV+/Scgn+ n = 9; PV+/Scgn- n

*Figure 6 continued on next page*

*Figure 6 continued*

= 8) from 0 to 80 Hz. Shaded areas show SEMs across neurons. (**E,F**) Mean phase histograms of striatal PV+/Scgn-(**E**) and PV+/Scgn+ (**F**) interneurons for cortical oscillations of 0.4–80 Hz during cortical activation. (**G**) Histogram showing the proportions of PV+/Scgn- (green) and PV+/Scgn+ (blue) neurons that were significantly locked in each frequency range of cortical oscillation during cortical activation. (**H**) Mean vector lengths calculated across all PV+/Scgn- (green) and PV+/Scgn+ (blue) neurons recorded during cortical activation (PV+/Scgn+ n = 11; PV+/Scgn- n = 7). Shaded areas show SEMs across neurons. (**A, B, E, F**, frequencies between 0–5 Hz are separated to allow for a wider color scale)

The following source data is available for figure 6:

**Source data 1.** Source data for *Figures 6C,D,G,H*.

Fisher Exact Test, p<0.001. *Figure 8—source data 1*). A complementary analysis showed that SPN somata that were apposed to at least one axonal bouton of a PV+ interneuron received a significantly greater number of appositions from the axons of PV+/Scgn- interneurons (n = 46 SPN somata) as compared to appositions from the axons of PV+/Scgn+ interneurons (n = 44 SPN somata; *Figure 8D*, Mann Whitney, p=0.009. *Figure 8—source data 1*). Taken together, these data not only show that the axons of both PV+/Scgn- interneurons and PV+/Scgn+ interneurons innervate SPN cell bodies, but also that the former interneuron population is more likely to do so and with more appositions per targeted SPN.

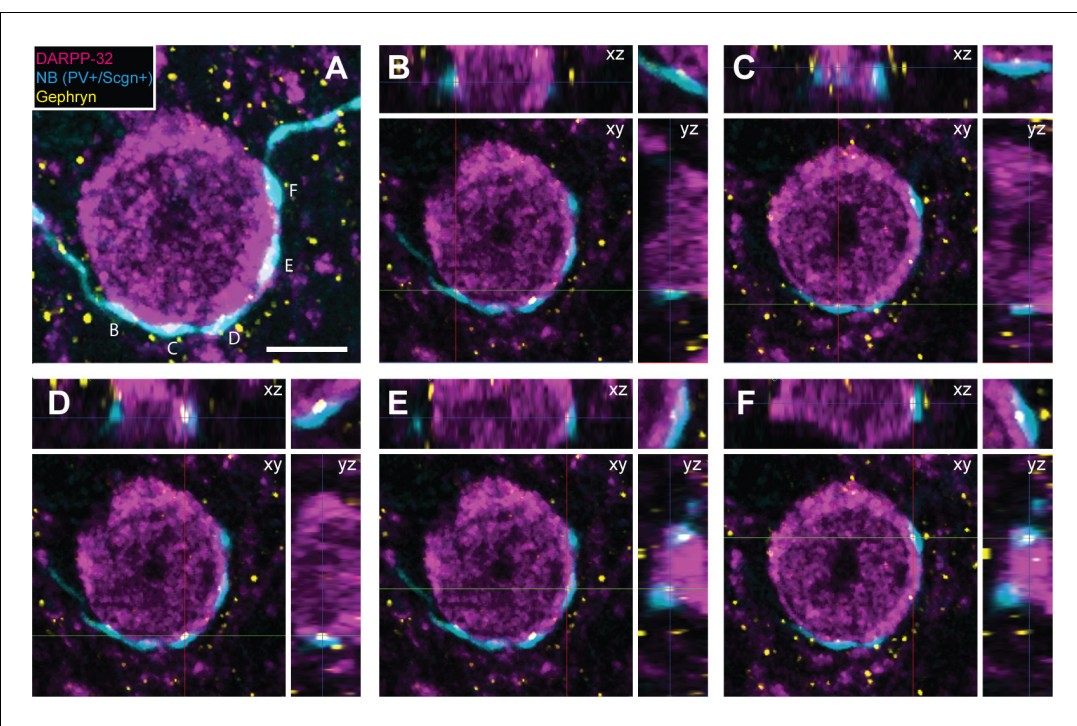

**Figure 7.** Appositions formed by the axons of PV+ interneurons with the somata of SPNs are associated with the post-synaptic marker gephyrin. (**A**) Confocal micrograph stack of an SPN cell body, labeled with DARPP-32 (purple), that is apposed by the boutons of a neurobiotin (NB)-labeled axon of a PV+/Scgn+ interneuron (blue). The section has also been labeled with an antibody against the post-synaptic structural protein gephyrin (yellow). Scale bar = 5 μm. (**B–F**), Single-plane confocal micrographs showing the xy axis and the corresponding orthogonal views; xz (top) and yz (right) of the 5 appositions labeled in **A**. Axes crossing points are marked by colored lines. The top right corner panel shows a magnification of the apposition in the xy axis. Note that, for each apposition, there is a punctate gephyrin signal located between the axonal bouton and the soma, indicating the presence of a putative GABAergic synapse.

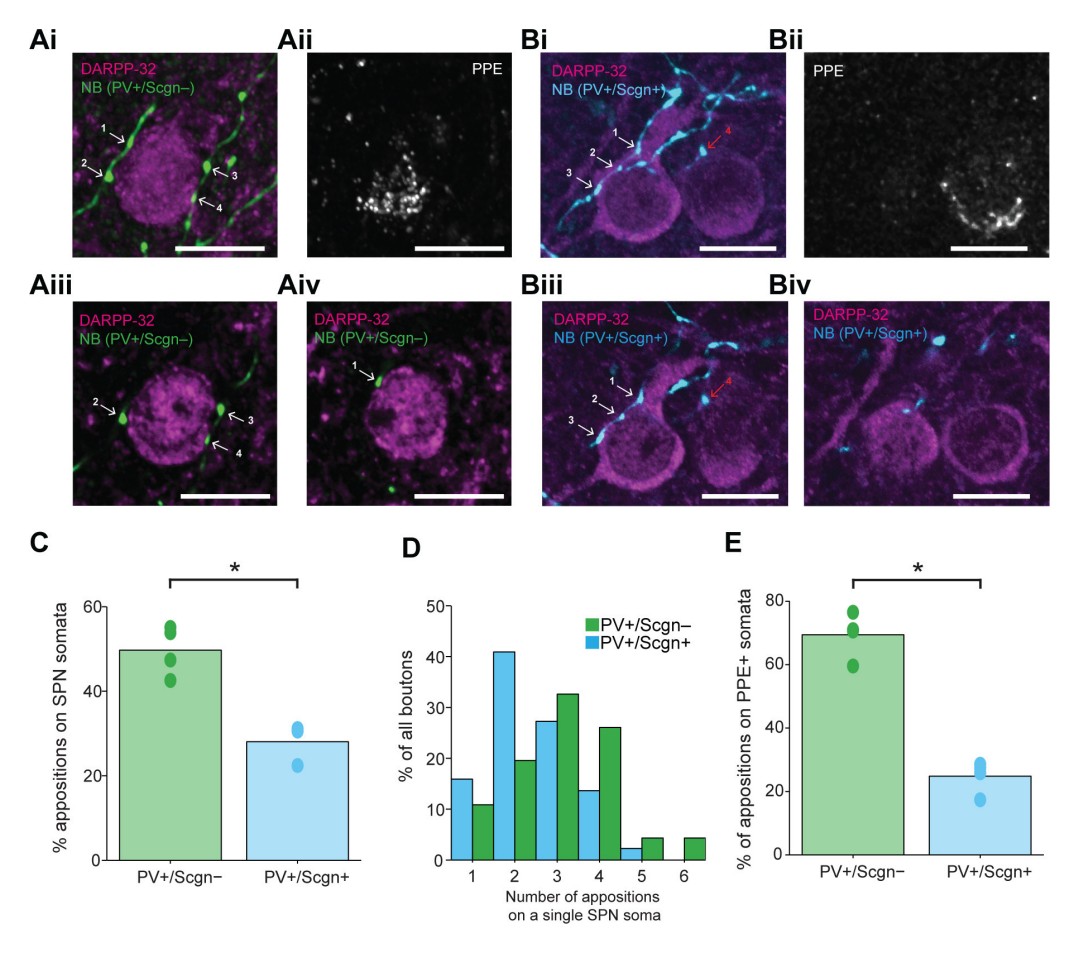

**Figure 8.** PV+/Scgn- and PV+/Scgn+ striatal interneurons selectively target the somata of SPNs in the direct or indirect pathway. (**Ai**) Confocal fluorescent micrograph stack of a neurobiotin (NB)-labeled axon of a PV+/Scgn- interneuron (green) targeting a SPN revealed with DARRP-32 (purple). The interneuron axon forms 4 appositions (numbered white arrows) with the SPN soma. (**Aii**) The SPN expresses preproenkephalin (PPE, white), indicating it is in the indirect pathway (iSPN). (**Aiii, Aiv**) Single-plane confocal micrographs verifying that each of the 4 boutons is closely apposed to the SPN soma. (**Bi**) Confocal fluorescent micrograph stack of the NB-labeled axon of a PV+/Scgn+ interneuron (blue) traversing close to two SPNs. The axon forms 3 appositions (white arrows) with the SPN soma on the left, and appears to form a single apposition (red arrow) with the SPN on right. (**Bii**) The right SPN expresses PPE (iSPN), while the left SPN does not, indicating it is in the direct pathway (dSPN). (**Biii, Biv**), Single-plane confocal micrographs verifying that boutons 1–3 are apposed to the soma of the dSPN, while bouton 4 is not directly apposed to the iSPN. (**C**) Quantitative analyses of the NB-labeled axonal boutons of PV+/Scgn- and PV+/Scgn+ interneurons revealed that the axons of PV+/Scgn- interneurons were more likely to be opposed to the somata of SPNs. (**D**) Histogram of the frequency of different numbers of appositions formed with an individual SPN soma for both types of interneuron. (**E**) The axons of PV+/Scgn- interneurons were more likely than the axons of PV+/Scgn+ interneurons to target the somata of PPE+ SPNs of the indirect pathway.(**A,B**, scale bars are 20 μm.)

The following source data is available for figure 8:

**Source data 1.** Source data for *Figure 8C,D,E*.

Fundamental to the conceptual organization of basal ganglia function is the separation of SPNs into two information streams, the so-called direct and indirect pathways, which have broadly antagonistic roles in the expression of behavior (*Gerfen and Surmeier, 2011*). Thus, we next examined whether there was any bias in the proportion of appositions formed between the PV+ interneuron axons and the somata of SPNs of the direct pathway (dSPNs) and indirect pathway (iSPNs). Somatic co-expression of DARPP-32 and preproenkephalin (PPE) was used to identify iSPNs, whereas those SPNs that did not express PPE were considered to be dSPNs (*Ellender et al., 2011*; *Lee et al., 1997*). The axons of individual PV+/Scgn- interneurons could readily be observed to form multiple

basket-like appositions around PPE+ iSPNs (*Figure 8A*), whereas the axons of PV+/Scgn+ interneurons appeared to more often target PPE- dSPNs (*Figure 8B*). Quantitative analysis confirmed that these two populations of PV+ interneurons significantly differed in their preferential targeting of the somata of iSPNs and dSPNs (Fisher Exact Test, p<0.0001). More specifically, the axonal boutons of PV+/Scgn- interneurons (n = 119 boutons from 4 interneurons) were more commonly apposed to iSPN somata than to dSPN somata (*Figure 8A,E*, *Figure 8—source data 1*). In contrast, the axons of PV+/Scgn+ interneurons (n = 238 boutons from 4 interneurons) preferentially targeted dSPN somata (*Figure 8B,E*, *Figure 8—source data 1*). In tissue sections containing the neurobiotin-labeled axons of PV+ interneurons, approximately half of all SPNs located within 30 μm (i.e. approximately 2 SPN cell body diameters) of an axonal bouton were iSPNs. This meant that the mean ratio of dSPNs to iSPNs in each section containing labeled interneuron axon was close to 1.0 (ratios of 1.04 ± 0.07 and 1.00 ± 0.12 for sections containing axons of PV+/Scgn+ or PV+/Scgn- interneurons, respectively). This indicates that the preferential targeting of dSPNs and iSPNs by the axons of PV+/Scgn+ and PV+/Scgn- interneurons, respectively, was not the result of a preferential distribution or enrichment of dSPNs or iSPNs in these specific tissue sections. Taken together, these results show that not only do PV+/Scgn- interneurons and PV+/Scgn+ interneurons target SPN somata to different extents, but also that these two types of interneuron preferentially innervate the somata of distinct populations of SPNs. By virtue of these biased connections, PV+/Scgn- interneurons might be better positioned than PV+/Scgn+ interneurons to selectively shape the activity of SPNs of the indirect pathway, whereas PV+/Scgn+ neurons might be better positioned to selectively influence SPNs of the direct pathway.

## PV+/Scgn- and PV+/Scgn+ interneurons differ in their temporal relationships with striatal projection neurons of the direct and indirect pathways

Striatal PV+ interneurons are often considered to provide 'feedforward' inhibition to SPNs (*Silberberg and Bolam, 2015*). Indeed, PV+ interneurons can rapidly integrate synchronized cortical inputs and thence curtail or delay SPN firing (*Gittis et al., 2010*; *Koos and Tepper, 1999*; *Mallet et al., 2005*; *Ramanathan et al., 2002*; *Sharott et al., 2012*; *Planert et al., 2010*). Our analysis of neurobiotin-labeled PV+/Scgn- and PV+/Scgn+ interneurons suggests these two cell types selectively innervate iSPNs and dSPNs, respectively. When our observations are placed within a framework of feedforward inhibition in vivo, one might predict that, first, the firing of PV+/Scgn- interneurons is more likely to precede the firing of iSPNs than dSPNs, and secondly, that the firing of PV+/Scgn+ interneurons is more likely to precede the firing of dSPNs than iSPNs.

To test these predictions, we compared the spike timing of identified SPNs (n = 48) and PV+ interneurons (n = 26) recorded in anesthetized rats during SWA (*Figure 9*). A subset of recorded and neurobiotin-labelled SPNs (n = 36 of 48) were tested for their expression of PPE, which led to the identification of 18 dSPNs (*Figure 9Ai*) and 18 iSPNs (*Figure 9Bi*). A subset of PV+ interneurons (n = 15 of 26) were tested for their co-expression of Scgn, which led to the identification of 8 PV+/Scgn- and 7 PV+/Scgn+ interneurons. As a common reference point for the temporal analysis of all striatal neuron firing, we used the peak of the slow oscillation (~1 Hz) present in the inverted striatal local field potential (iLFP) that was simultaneously recorded with the single-unit activity (*Figure 9Aii, Bii*). The slow oscillation in the iLFP is of particular relevance because it is a proxy signal for the cortically-driven synchronized 'up states' of many SPNs in the vicinity of the electrode (*Goto and O'Donnell, 2001*; *Stern et al., 1998*). For each striatal neuron, we calculated histograms of the spike times and iLFP amplitudes across the iLFP peak in 100 time bins, irrespective of the length of the individual cycle (*Figure 9A–C*); this normalization procedure ensured that the variable durations of the slow oscillation components (*Nakamura et al., 2014*) and thus, variable 'peak lengths', did not confound the analysis.

When all recorded SPNs were grouped together for analysis, their maximal firing probability occurred before the center of the iLFP peak, and their firing probability 'profile' extensively overlapped with that of all PV+ interneurons when analyzed together (*Figure 9Ci*, *Figure 9—source data 1*). These observations agree with previous reports (*Mallet et al., 2005*), and are further supported by the fact that the median firing times of the SPN and PV+ interneurons were not different (Mann Whitney, p=0.55. *Figure 9D*, *Figure 9—source data 1*). However, when the subsets of identified dSPNs and iSPNs were separately analyzed, their firing probability profiles only partly

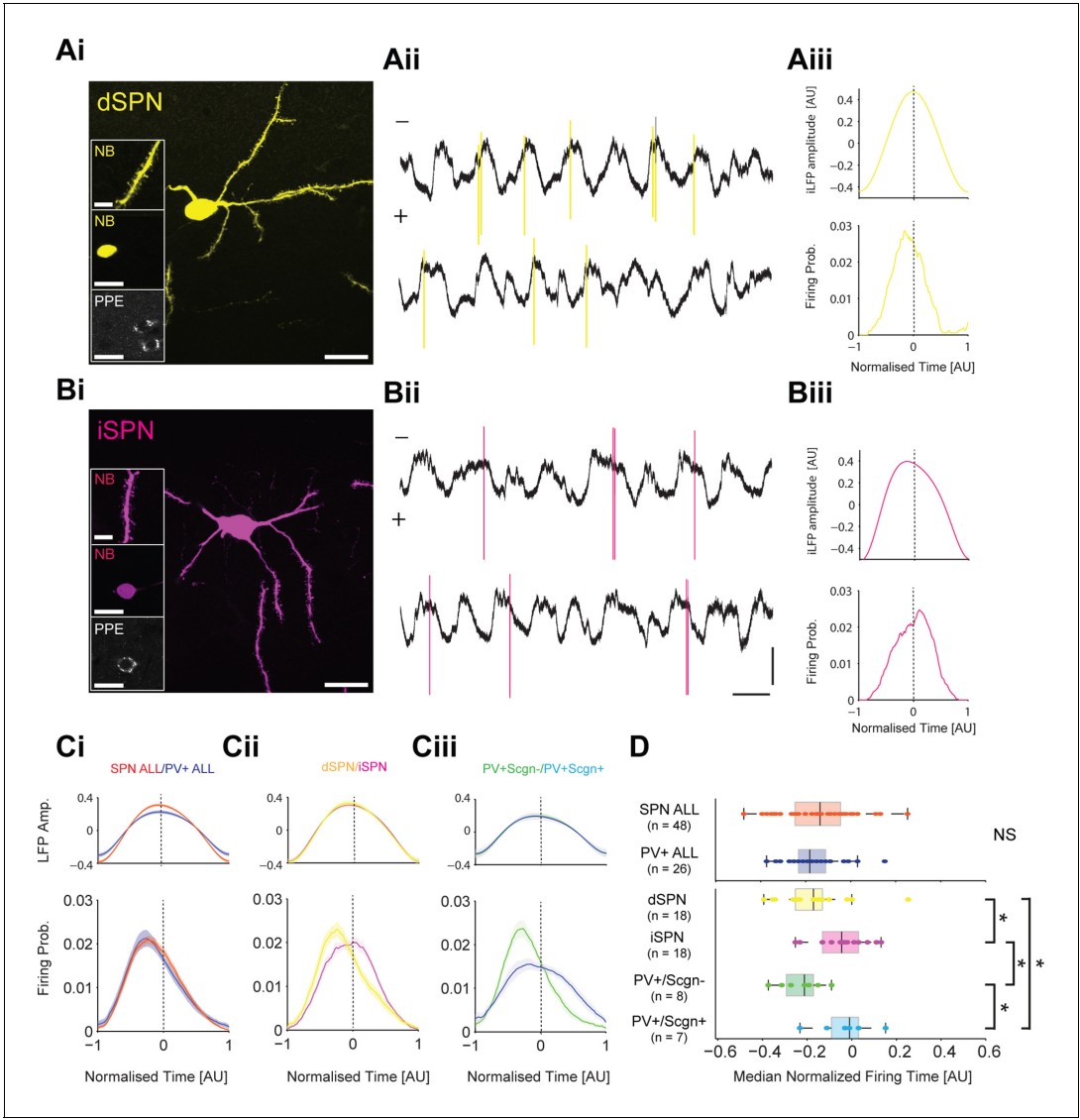

**Figure 9.** PV+/Scgn- and PV+/Scgn+ interneurons differ in their temporal relationships with spiny projection neurons of the direct and indirect pathways. (**Ai**) Striatal projection neuron (SPN), juxtacellularly labelled with neurobiotin (NB) and identified by its densely spiny dendrites (top inset). This neuron's soma does not express immunoreactvity for PPE (middle and bottom insets), identifying it as a direct pathway SPN (dSPN). (**Aii**) Two 10 s epochs of the inverted local field potential (iLFP) that were simultaneously recorded with single-unit activity in the striatum (all from same glass electrode). The identified dSPN tended to fire action potentials (yellow) just before and around the peaks of the iLFP. (**Aiii**) Histograms of iLFP amplitude (top) and dSPN spike firing (below), confirming that the dSPN fires most often before the center of the iLFP peak. (**Bi**) Juxtacellularly-labelled indirect pathway SPN (iSPN), identified by its somatic expression of PPE. (**Bii, Biii**) The iSPN tended to fire action potentials (magenta) just after and around the peaks of the iLFP (Bii), which is confirmed by the histograms of iLFP amplitude and spike firing (**Biii**). (**C**) Mean iLFP histograms and spike-firing probability histograms for all SPNs and all PV+ interneurons recorded (i), for identified dSPNs and iSPNs (ii), and for identified PV+/Scgn- and PV+/Scgn+ interneurons (iii). Note that, when considered as whole populations, SPNs and PV+ interneurons fire at similar times with respect to the iLFP, but also that firing times diverge when the SPN and PV+ interneuron populations are divided according to their dichotomous molecular identities. (**D**) The median firing times of all SPN and all PV+ interneurons are not significantly different (upper plot). When the subpopulations are analyzed (lower plots), dSPNs fire significantly earlier than iSPNs, and PV+/Scgn- interneurons fire significantly earlier that PV+/Scgn+ interneurons. Furthermore, PV+/Scgn- interneurons fire significantly earlier than iSPNs, but not dSPNs. Note also that PV+/Scgn+ interneurons fire significantly later than dSPNs (Kruskal Wallis ANOVA on rank [p=0.0006] with post-hoc Dunn tests). AU, arbitrary units. (**Ai, Bi**, Scale bars are 20 μM, except for those in dendrite images, which are 5 μM **Aii, Bii**, Vertical scale bars for unit activity and iLFPs are 1 mV; Horizontal scale bars are 1 s).

The following source data is available for figure 9:

**Source data 1.** Source data for *Figure 9C, D*.

overlapped; dSPNs tended to fire before the center of the iLFP peak whereas iSPNs fired on or slightly after the center (*Figure 9Cii*). Accordingly the median firing time of dSPNs significantly preceded that of the iSPNs (*Figure 9D*, *Figure 9—source data 1*). To give further context to this spike timing difference, we separately analyzed the firing of the identified PV+/Scgn- and PV+/Scgn+ interneurons. The firing probability profiles of the two types of PV+ interneuron diverged substantially; the profile of PV+/Scgn- interneurons was sharp with a maximum well before the center of the iLFP peak, whereas the profile of PV+/Scgn+ interneurons was broad and maximal around the center (*Figure 9Ciii*). Analysis of median firing times confirmed that PV+/Scgn- interneurons fired significantly earlier than PV+/Scgn+ interneurons (*Figure 9D*, *Figure 9—source data 1*), thus highlighting further disparities in the in vivo firing properties of these two types of PV+ interneuron. When the firing times of PV+/Scgn- interneurons, PV+/Scgn+ interneurons, dSPNs and iSPNs were compared (*Figure 9D*, *Figure 9—source data 1*), it was evident that PV+/Scgn- interneurons fired significantly earlier than iSPNs but not dSPNs, whereas PV+/Scgn+ interneurons did not fire before iSPNs or dSPNs (their firing was significantly delayed as compared to dSPNs). There were no significant differences in the peak times, lengths or amplitudes of the iLFP peaks between the different neuron groups (Kruskal-Wallis ANOVA with post-hoc Dunn tests), suggesting that the differences in firing times of the distinct neuron types were not due to systematic biases in the LFPs recorded with each population.

These electrophysiological data verify the first prediction above with respect to possible substrates for feedforward inhibition, that is, the firing of PV+/Scgn- interneurons is indeed more likely to precede the firing of iSPNs than dSPNs. This result is consistent with our anatomical data showing that, in the context of targeting SPN somata, PV+/Scgn- interneurons exhibit a considerable bias towards innervating iSPNs. The firing of PV+/Scgn+ interneuron firing with respect to dSPN firing did not validate the second prediction; the spike timings of this type of interneuron are not consistent with a powerful feedforward inhibitory connection to either type of SPN. Again, this result resonates with our anatomical data showing that, although PV+/Scgn+ interneuron preferentially target dSPNs, these interneurons form fewer putative somatic synapses and fewer appositions on a given SPN soma. Taken together, these observations suggest that the diverse properties of PV+/Scgn- and PV+/Scgn+ interneurons position them to fulfil different roles in the striatal circuit.

## Hierarchical clustering of striatal interneurons using electrophysiological parameters is highly correlated with their molecular identities

The analyses above show that PV+/Scgn- and PV+/Scgn+ interneurons differ in several of their electrophysiological properties, suggesting they are discrete cell types. These analyses were necessarily based on comparisons of two sets of PV+ interneurons that were first divided according to their selective expression of Scgn. This raises the issue of whether PV+ interneurons can be segregated into two or more discrete groups on the basis of their electrophysiological properties alone. If they can be segregated, this raises the further issue of whether the discrete interneuron groups differ with respect to their expression of Scgn. To address these issues, we performed unsupervised hierarchical cluster analyses of the electrophysiological parameters of PV+ interneurons (omitting information on whether or not they expressed Scgn) together with those of cholinergic interneurons and GABAergic interneurons that co-express nitric oxide synthase (NOS) and neuropeptide Y (NPY). We included ChAT+ interneurons and NOS+/NPY+ interneurons because they are widely accepted to be discrete cell types (*Tepper and Bolam, 2004*) and, as such, can be used as comparators; if PV+/Scgn- and PV+/Scgn+ interneurons can be distinguished to the same degree as ChAT+ and NOS+/NPY+ interneurons can be distinguished (from each other and/or from the subpopulations of PV+ interneurons), then this would support the notion that PV+/Scgn- and PV+/Scgn+ interneurons are discrete cell types. Another key advantage of including ChAT+ and NOS+/NPY+ interneurons is that they facilitated the unbiased selection of the electrophysiological parameters to analyze; parameters were thus selected according to their utility for segregating one or more of the populations of ChAT+ interneurons, PV+ interneurons (as a whole) and NOS+/NPY+ interneurons, rather than their ability to distinguish PV+/Scgn- from PV+/Scgn+ interneurons per se.

Eighty three percent of the cholinergic interneurons and all of the NOS+/NPY+ interneurons used here have been reported in previous papers (*Sharott et al., 2012*; *Doig et al., 2014*). Because the firing rates and patterns of striatal interneurons varies considerably between SWA and cortical

activation (*Sharott et al., 2012*), we performed separate cluster analyses for parameters recorded in each brain state. For activity during SWA, we analyzed a total of 65 interneurons; 36 ChAT+, 12 NOS+/NPY+ and 17 PV+ interneurons. Three measures of interneuron firing regularity/pattern (Log ISI 10, CV ISI and CV2 ratio; see Materials and methods) and 4 measures of interneuron locking to population oscillations in the ECoG (LFP peak, SWA Vec., Spin. Vec and Gam. Vec; see Materials and methods) were used for clustering (*Figure 10Ai*, *Figure 10—source data 1*). When these 7 parameters were analyzed across all interneuron populations, 5 significant clusters emerged (*Figure 10B*). After assignment of molecular identities, it was evident that two of these clusters were predominantly composed of cholinergic interneurons (*Figure 10B*). The three remaining clusters were composed of a clear majority of PV+/Scgn-, PV+/Scgn+ or NOS+/NPY+ interneurons. The PV+/Scgn- and PV+/Scgn+ interneurons were therefore segregated to a similar degree as the ChAT+ and NOS+/NPY+ interneurons. When the ChAT+ interneurons were removed and the analysis repeated, the GABAergic interneurons segregated into 3 significant clusters with a slightly improved clustering of the NOS+/NPY+ interneurons and a similar separation of the two PV+ populations to the larger analysis (*Figure 10C*). With the removal of NOS+/NPY+ interneurons, the segregation of PV+/Scgn- and PV+/Scgn- interneurons was largely maintained (*Figure 10D*). When only the 4 measures of interneuron locking were used in the analysis, PV+ interneurons were segregated into two significant clusters with >85% correlation with Scgn expression (*Figure 10D*). This could reflect the relatively large influence of cortical oscillations on the firing patterns of striatal GABAergic interneurons in this brain state (*Sharott et al, 2012*).

For activity during cortical activation, we analyzed a total of 48 interneurons; 24 ChAT+, 6 NOS+/NPY+ and 18 PV+ interneurons. Six measures of interneuron firing regularity/pattern (Log ISI 10, Log ISI 50, Log ISI 85, mean firing rate, CV2 ratio and CV2 mean; see Materials and methods) were used for clustering (*Figure 10Aii*, *Figure 10—source data 1*). When these 6 parameters were analyzed across all interneuron populations, 4 significant clusters emerged (*Figure 10F*). After assignment of molecular identities, it was evident that these clusters were predominantly composed of cholinergic interneurons, NOS+/NPY+ interneurons, PV+/Scgn- interneurons or PV+/Scgn+ interneurons (*Figure 10F*), thus establishing the value of the parameters used. When the ChAT+ interneurons were removed and the analysis repeated, the NOS+/NPY+, PV+/Scgn- and PV+/Scgn+ interneurons remained significantly segregated to similar degrees (*Figure 10G*). These data show that, during cortical activation, PV+/Scgn+ and PV+/Scgn- interneurons are as distinct from each other as either subpopulation is from ChAT+ or NOS+/NPY+ interneurons. When the analysis was repeated after removal of NOS+/NPY+ interneurons, 3 significant clusters emerged, one of which was composed entirely of PV+/Scgn+ interneurons (*Figure 10H*). Interestingly, the remaining PV+ interneurons, which were mostly Scgn-, were divided into two clusters. This concurs with our ISI histogram correlation analysis (*Figure 5*) that indicated that the firing patterns of PV+/Scgn- interneurons are more variable than those of PV+/Scgn+ interneurons.

In summary, these unsupervised cluster analyses show not only that PV+ interneurons can be segregated into two discrete groups on the basis of their electrophysiological properties alone (in two brain states), but also that these discrete interneuron groups differ with respect to their Scgn expression. Taken together, these data further support the notion that rat PV+ interneurons are comprised of two main subpopulations, and that selective expression of Scgn is a robust and useful metric for distinguishing between them.

## Discussion

Striatal PV-expressing interneurons have generally been regarded as a single population or cell type. Here we demonstrate that, in rats and primates, co-expression (or lack thereof) of a second calcium-binding protein, secretagogin, divides PV+ interneurons into two functionally-specialized subpopulations by virtue of their different structural, topographical and physiological properties.

Neurons of the same cell type deliver identical neuroactive substances to a matching range of postsynaptic targets in the same temporal patterns (*Somogyi, 2010*). These properties can be defined using molecular, structural and physiological features (*Petilla Interneuron Nomenclature et al., 2008*). Exemplars of the utility of such a multi-level definition of GABAergic cell type include the discrimination of myriad types of hippocampal interneuron (*Klausberger and Somogyi, 2008*) and the cellular dichotomy recently revealed in the external globus pallidus (*Abdi et al., 2015*;

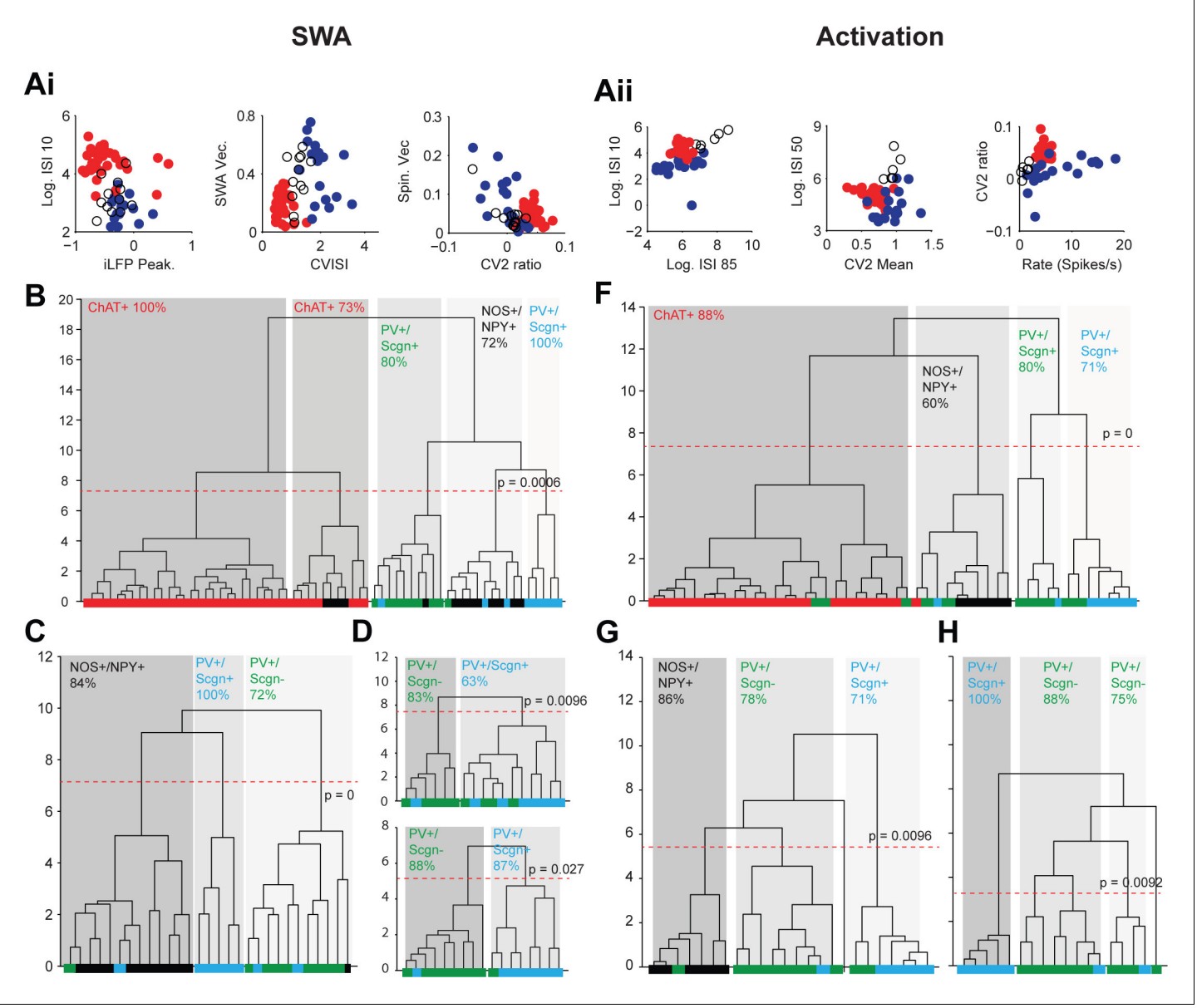

**Figure 10.** Unsupervised hierarchical clustering of electrophysiological parameters segregate PV+/Scgn+ interneurons from established striatal interneuron types. (A) Scatter plots showing the values of 6 electrophysiological parameters used in the cluster analysis of juxtacellularly-labelled ChAT+ (red), GABAergic NOS+/NPY+ (black circles) and PV+ (dark blue) interneurons recorded during SWA (Ai) and cortical activation (Aii). See Materials and ethods for definitions of each parameter. Each variable separates two or more of the interneuron groups. (B) Dendrogram derived from 7D-cluster analysis using Ward's method with a squared Euclidian distance measure to classify 65 striatal interneurons recorded during SWA using the parameters in Ai and one other (ECoG gamma vector length). The x-axis represents individual cells (ChAT+ in red, NOS+/NPY+ in black, PV+/Scgn- in green, PV+/Scgn+ in light blue), the y-axis represents average linkage distance between neurons, where longer distance represents greater dissimilarity. The dotted redline represents the threshold for separating clusters, which are highlighted by grey boxes, together with the p-value for this threshold. Five clusters are formed, two made up mostly of ChAT+ interneurons and the other three of different types of GABAergic interneurons. (C, D) The same analysis run only on the GABAergic interneurons (C) and only on PV+ interneurons (D). (C) The three significant clusters correspond to the three different molecular marker combinations with >70% accuracy. (D) *Upper*, in the 7D space, the two significant clusters for PV+ interneurons are roughly segregated according to Scgn expression. *Lower*, 4D-cluster analysis using only parameters related to population locking. The two significant clusters are almost completely predicted by Scgn expression. (E) Dendrogram of 6D cluster analysis of 48 interneurons recorded during cortical activation using the parameters in Aii. 4 clusters are significantly segregated, each with a clear majority of cells with a single molecular identity. (F) Analysis of only GABAergic interneurons leads to 3 significant clusters highly correlated with the 3 interneuron types. (G) Analysis of only PV+ interneurons led to 3 significant clusters, 2 of which were predominantly comprised of PV+/Scgn- interneurons. The third cluster, which had the widest segregation, was comprised only of PV+/Scgn+ interneurons.

*Figure 10 continued on next page*

*Figure 10 continued*

The following source data is available for figure 10:

**Source data 1.** Source data for *Figure 10*.

*Mallet et al., 2012*). In both of these cases, a specific combination of molecular markers can be used as the primary or sufficient indicators of the unique set of structural and physiological properties. Combinations of markers have also proved useful for parsing some types of GABAergic interneuron in the rodent striatum. For example, interneurons expressing NPY can be separated according to whether or not they also express NOS and/or somatostatin; those expressing NPY, but not NOS, have a completely distinctive anatomical and physiological phenotype (*Ibanez-Sandoval et al., 2011*) and specific role in the microcircuit (*English et al., 2012*) as compared to NPY+/NOS+ interneurons. In contrast, striatal PV+ interneurons have generally been treated as a homogenous population. The findings presented here describing the molecular profiles, distribution, structure and firing of striatal PV+ interneurons converge to suggest that striatal PV+ interneurons in rats and monkeys comprise at least two discrete cell types. The unsupervised cluster analysis of PV+ interneuron firing during cortical activation suggested this heterogeneity may extend further.

We compared PV and Scgn expression across three species in which the amino acid sequence of Scgn is highly conserved (*Gartner et al., 2007*; *Pruitt et al., 2014*; *Zierhut et al., 2005*). While PV+/Scgn+ interneurons constituted around 30% and 75% of all PV+ interneurons in rats and primates respectively, Scgn was only expressed by a small minority of PV+ interneurons (<1%) in mice. Because Scgn-expressing neurons could be observed in other brain regions of mice, the scarcity of PV+/Scgn+ interneurons in mouse striatum was thus unlikely to be due to a technical/detection issue. The paucity of Scgn co-expression in mouse PV+ interneurons suggests the molecular diversity we elucidate here does not tally with that defined by the selective, but more widespread, expression of serotonin receptor subunit 3A (*Munoz-Manchado et al., 2016*). However, it would be important to test in the future whether Scgn expression in rat and primate striatum correlates with expression of this receptor subunit. It is possible that mouse striatum still contains a sub-population of PV+ interneurons that have the same anatomical and physiological properties as the PV+/Scgn+ interneurons in rats. Even if this scenario were true, however, the fact that such mouse interneurons lack Scgn suggests they would handle calcium differently compared to rat/primate interneurons that co-express Scgn. Moreover, given the increasing use of genetically-altered mice for identifying and manipulating specific neuronal populations based on their selective gene expression, the finding that a combination of molecular markers is so prominent in the rat and primate but absent in the mouse striatum is noteworthy. Indeed, further study of the functional properties of PV+/Scgn- and PV+/Scgn+ interneurons using, for example, cell-type-specific optical or pharmacogenetic methods, would likely require a double-transgenic rat, the generation of which would present a considerable technical challenge. However, this does not alter the likelihood that, although many elements of striatal circuits are likely conserved across mice, rats and primates, a focus on mice as a model organism could potentially miss important substrates of interneuron diversity that are conserved along different phylogenetic paths.

The corticostriatal projection is topographically organized and (depending on the precise cortical areas of origin) partly overlapping, with different striatal territories receiving specific combinations of cortical afferents that impart diverse functionality. As the predominant excitatory (glutamatergic) input to striatal PV+ interneurons is from cerebral cortex (*Lapper et al., 1992*; *Ramanathan et al., 2002*), the origin of these inputs is likely to be particularly important in defining their function. In rat, PV+/Scgn+ interneurons were distributed across the entire striatum, but their density dramatically increased in caudal aspects (such that they outnumbered PV+/Scgn- interneurons by >10 to 1). These caudal striatal territories receive converging inputs from, in particular, most areas of prefrontal (*Hoover and Vertes, 2011*; *Mailly et al., 2013*), entorhinal (*McGeorge and Faull, 1989*), prelimbic (*Mailly et al., 2013*), visual (*Faull et al., 1986*) and auditory cortical areas (*McGeorge and Faull, 1989*; *Xiong et al., 2015*). In contrast, PV+/Scgn- interneurons were evenly distributed throughout striatum, with the exception of caudal territories where they were sparse. Thus, corticostriatal neurons in areas of rat cortex that heavily innervate caudal striatum could gain privileged access to PV+/

Scgn+ interneurons. On the other hand, in rostral and central territories of striatum, which receive denser and more widespread inputs from somatosensory and motor cortical areas, PV+/Scgn- interneurons outnumbered PV+/Scgn+ interneurons by at least 3:1. Thus, corticostriatal neurons in 'somato-motor' areas of rat cortex are particularly well positioned to engage PV+/Scgn- interneurons.

Primate corticostriatal projections differ in several ways to those of rats and mice. The somatosensory/motor cortical areas in primates comprise a far smaller fraction of the entire cortical mantle than in rodents, particularly in relation to prefrontal, association and visual cortical areas (*Brodmann, 1909*; *Buckner and Krienen, 2013*). In line with this change in relative cortical volume, combinations of afferents from prefrontal, association and visual cortices of primates innervate large areas of the caudate and putamen across most of their rostro-caudal axes (*Calzavara et al., 2007*; *Haber and Knutson, 2010*; *Yeterian and Pandya, 1998*), whereas corticostriatal neurons in motor and somatosensory cortical areas primarily target the lateral putamen (*Flaherty and Graybiel, 1991*; *Inase et al., 1996*; *Parent and Parent, 2006*; *Takada et al., 1998*). These differences in the sources and sizes of corticostriatal innervation are partly correlated with the relative densities and distributions of the two populations of striatal PV+ interneurons in the primate. Unlike in the rat, PV+/Scgn+ interneurons in primates were three times as dense as PV+/Scgn- interneurons across the entire rostro-caudal extent of striatum. In further contrast to the rat, PV+/Scgn- interneurons were clustered in lateral aspects of the putamen. Thus, the relative switch in prevalence of the two PV+ interneuron populations in the two species is roughly in line with the relative difference in prevalence of somatosensory/motor corticostriatal projections versus prefrontal, associative, visual and auditory corticostriatal projections. There was another notable similarity between rat and macaque, namely that the density of PV+/Scgn+ interneurons increased in caudal territories. Both rat caudal striatum (*Faull et al., 1986*) and primate caudal caudate receive dense inputs from visual cortical areas (*Gattass et al., 2014*), which could possibly explain this shared relationship.

In hippocampus, PV is expressed by at least four distinct types of GABAergic interneuron (axo-axonic, basket, bistratified and O-LM cells) that selectively innervate different compartments of target cells (*Klausberger and Somogyi, 2008*). Approximately 50% of PV-immunoreactive terminals in the striatum form symmetric synapses with putative SPN somata (*Bennett and Bolam, 1994*; *Kita et al., 1990*). While some of these terminals could originate from neurons in external globus pallidus (*Bevan et al., 1998*; *Mallet et al., 2012*), a much larger proportion are likely to originate from PV+ interneurons (*Koos and Tepper, 1999*; *Kubota and Kawaguchi, 2000*). Thus, PV+ interneurons in striatum have generally been thought to play a similar circuit role to basket cells in cortex. We found that, as compared to outputs of PV+/Scgn+ interneurons, the PV+/Scgn- interneurons formed almost twice as many putative somatic synapses and formed more appositions on a given SPN soma. From our experiments, we can only speculate on the identities of other targets innervated by PV+/Scgn+ and PV+/Scgn- interneurons. While it seems likely that these interneurons form some synapses with SPN dendrites, we cannot rule out that the somata and/or dendrites of interneurons are also targeted. Whatever their targets, our results suggest that different subpopulations of striatal PV+ interneuron could also underlie the considerable variability in the cellular compartments targeted by individual PV+ interneuron axons (*Kubota and Kawaguchi, 2000*).

The targeting of SPN somata is an important factor in the ability of PV+ interneurons to powerfully inhibit SPNs (*Gittis et al., 2010*; *Planert et al., 2010*). Bursts of spikes in PV+ interneurons can considerably delay or even prevent SPN spiking, and inhibition mediated by PV+ interneurons is considerably stronger than that mediated by other GABAergic interneurons (*Gittis et al., 2010*; *Koos and Tepper, 1999*; *Planert et al., 2010*). We demonstrate that the axon of a single PV+ interneuron most commonly forms 2–4 putative synapses with a single SPN soma. Thus, in addition to several PV+ interneurons forming basket-like contacts with an SPN soma, our results indicate that an axon from an individual PV+ interneuron can form sufficient synaptic contacts to powerfully inhibit a single SPN.

In vitro electrophysiological recordings in mice have suggested that PV+ interneurons provide either approximately equal inhibition to dSPNs and iSPNs, or have a slightly greater influence on dSPNs (*Gittis et al., 2010*; *Planert et al., 2010*). Our findings suggest that the innervation of SPNs by PV+ interneurons in the rat and primate is considerably more complex. In the context of targeting SPN somata, we found that PV+/Scgn- and PV+/Scgn+ interneurons had a considerable bias towards innervating iSPNs and dSPNs, respectively. The terminals of corticostriatal neurons show

little selectivity for the two SPN populations (*Doig et al., 2010*; *Guo et al., 2015*; *Kress et al., 2013*; *Wall et al., 2013*). The biased innervation of SPNs by two different types of PV+ interneuron may, therefore, provide a novel mechanism through which corticostriatal neurons can selectively influence one or the other output pathway. The utility of this mechanism would be further enhanced by the high sensitivity of PV+ interneurons to cortical input (*Gittis et al., 2010*; *Mallet et al., 2005*) and/or by any differences in the temporal organisation of PV+/Scgn- and PV/Scgn+ interneuron activities. We observed that the firing of these two types of PV+ interneuron was temporally distinct with respect to the slow oscillations present in the striatum during SWA; PV+/Scgn- interneurons fired earlier than PV+/Scgn+ interneurons. Selective innervation of iSPNs by PV+/Scgn- interneurons might thus explain why the firing of iSPNs was delayed compared to that of dSPNs. In contrast, the relative timing of PV+/Scgn+ interneuron firing and dSPN firing would not be consistent with a powerful feedforward inhibitory circuit. In the future, in vitro recordings of synaptically-connected pairs of PV+ interneurons and SPNs should provide further insights into the predicted functional impact of the selective targeting of the somata of dSPNs and iSPNs.

Innervation of the somatic compartments of pyramidal neurons by some types of cortical PV+ interneurons is a key factor contributing to the role of these interneurons in the generation of network oscillations (*Freund and Katona, 2007*; *Klausberger and Somogyi, 2008*). The strong temporal relationships between the firing of PV+ interneurons or FSIs to cortical oscillations is also well described in striatum (*Berke, 2004*, *2009*; *Sharott et al., 2012*, *2009*). Thus, it was notable that PV+/Scgn- and PV+/Scgn+ interneurons displayed differences in the incidence, strengths and timings of their phase-locked firing to different cortical rhythms. This could be the result of these two interneuron populations receiving different cortical inputs with different oscillatory signatures. Alternatively, variance in the intrinsic properties of these interneurons could result in a different output in response to the same oscillatory inputs. Whatever the underlying mechanism, our results highlight that the physiological differences of PV+/Scgn- and PV+/Scgn+ interneurons extend to their disparate engagement in behaviorally-relevant network oscillations in the corticostriatal circuit. In addition, the firing pattern within the PV+/Scgn+ interneurons was as similar to each other as to that of cholinergic interneurons, which have relatively homogenous firing patterns (*Sharott et al., 2012*). In contrast, PV+/Scgn- interneurons displayed considerable variability in their firing patterns, which could indicate that they represent more than one subtype of interneuron.

Overall, our findings suggest that PV+/Scgn- interneurons and PV+/Scgn+ interneurons play different roles in the feedforward inhibition of striatal outputs, which are in turn likely to underpin specialized contributions to the control of behavior. A recent study in mice demonstrated feedforward inhibition of SPNs in striosomes, but not matrix compartment, of rostro-medial striatum by afferents from prelimbic cortex (*Friedman et al., 2015*). Optogenetic manipulation of this corticostriatal/inhibitory-interneuron pathway specifically altered performance in cost/benefit decision making, illustrating that inhibitory interneurons can enable cortical afferents from a single area to selectively mediate a subset of striatal outputs (*Friedman et al., 2015*). We propose that PV+/Scgn+ interneurons may instantiate a similar mechanism, whereby associative, visual and auditory cortical inputs can selectively inhibit dSPNs. Consistent with this idea, the caudal striatum of rats has been shown to mediate the association of auditory stimuli with rewarding actions (*Xiong et al., 2015*). The proliferation of PV+/Scgn+ interneurons in the macaque striatum could reflect the increase in such decision-making capabilities required by the corticostriatal system of primates.

## Materials and methods

### Indirect immunofluorescence of rat and mouse tissue

The experimental procedures described below were carried out using 36 adult (3–4 months old, 280–350 g) male Sprague Dawley rats (Charles River) and 7 adult (3–4 months old) C57Bl/6J male mice (Charles River) in accordance with the Animals (Scientific Procedures) Act, 1986 (UK). After being deeply anaesthetized using isoflurane (4% v/v in oxygen), each rat was given a lethal dose of pentobarbitone (1.3g/kg; i.p.) and transcardially perfused with approximately 50 ml of 0.05 M phosphate-buffered saline, pH7.4 (PBS), followed by 300 ml of fixative (4% w/v paraformaldehyde with 0.1% w/v glutaraldehyde in 0.1 M phosphate buffer, pH7.4 (PB)). This was followed by a third perfusion of approximately 200 ml of fixative (4% w/v paraformaldehyde in PB). Mice were deeply

anesthetized with pentobarbitone and perfused transcardially using 20 ml of PBS, followed by 20 ml of fixative (4% w/v paraformaldehyde in 0.1 M PB). For both species, once the brain was removed, the tissue was post-fixed in this solution for 24 hr at 4°C. Using a vibrating-blade microtome (Leica VT1000S), 50 µm-thick coronal sections containing the striatum as identified using either a rat or mouse brain atlas (*Franklin and Paxinos, 2008*; *Paxinos and Watson, 2007*) were cut and collected in a one in four series for immunofluorescence processing.

Sections were washed with PBS (four 10 min washes) and pre-incubated for 2 hr in a solution consisting of 10% v/v normal donkey serum (NDS) and 0.3% v/v Triton X-100 in PBS. After further washing using PBS, sections were incubated overnight at room temperature in a solution of 0.3 %v/v Triton X-100 in PBS containing primary antibodies. Refer to *Table 1* for details, source, and dilutions of antibodies used. After incubation in primary antibodies, sections were washed in PBS and incubated overnight at room temperature in Triton-PBS which contained a mixture of secondary antibodies (all raised in donkey) which were conjugated to the following fluorophores: DyLight 649 (1:500; Jackson ImmunoResearch Laboratories); Cy3 (1:1000; Jackson ImmunoResearch Laboratories); Alex-aFluor-488 (1:500; Invitrogen); or AMCA (1:250 dilution; Jackson ImmunoResearch Laboratories). To ensure minimal cross-reactivity, these antibodies were cross adsorbed by the manufacturers. After further washing in PBS, sections were mounted on glass slides using fluorescence mounting medium (Vectashield; Vector Laboratories), followed by the addition of a coverslip.

## Sampling and cell-counting strategies in rat and mouse tissue

Using a series of partly overlapping, complementary immunofluorescence protocols, striatal neurons were tested for their combinatorial expression of some of the molecular markers in *Table 1*. A version of design-based stereology, the 'modified optical fractionator' (*Dodson et al., 2015*; *Abdi et al., 2015*), was used to generate unbiased cell counts, determine the relative expression of molecular markers, and map distributions of striatal interneurons. In all procedures performed, the accuracy of these estimates was ensured by performing counts throughout the entirety of the striatum within a given rostro-caudal plane. This allowed for an accurate description of their distribution within the striatum. The analysis of dorsal striatal expression of specific molecular markers was performed using 13 coronal striatal planes in the rat (*Paxinos and Watson, 2007*). For protocols performed using mouse striatum, 9 coronal planes were used (*Franklin and Paxinos, 2008*). As our aim was to investigate the distribution of interneurons across specific coronal planes in a quantitatively and statistically robust manner, chosen sections from all animals were matched with one another such that sections representing the same distance from Bregma (±50 µm) were chosen. This ensured that the results from all animals studied could be averaged to produce more accurate results regarding the number of counted neurons within a given coronal plane.

Once the chosen striatal coronal planes were identified and the immunofluorescence protocol carried out, the dorsal striatum was delineated using a Zeiss Imager M2 epifluorescence microscope (Carl Zeiss, AxioImager.M2) equipped with a 20X (Numerical Aperture = 0.8) objective and StereoInvestigator v9.0 software (MBF Biosciences). In order to image each fluorescence channel, the following sets of filter cubes were used: AMCA (excitation 299–392 nm, beamsplitter 395 nm, emission 420–470 nm); AlexaFluor-488 (excitation 450–490 nm, beamsplitter 495 nm, emission 500–550 nm); Cy3 (excitation 532–558 nm, beamsplitter 570 nm, emission 570–640 nm); and DyLight 649 (excitation 625–655 nm, beamsplitter 660 nm, emission 665–715 nm). Imaging was subsequently performed by capturing a series of completely tessellated, z-stacked images (each 1 µm thick) at depths from 2 to 12 µm from the upper surface of each section at the level of the striatum (thereby defining a 10 µm-thick optical disector). As counts were performed across the entirety of the dorsal striatum within a given rostro-caudal plane, the grid size and counting frame were set to the same size of 420 x 320 µm. In order to calculate the coefficient of error associated with each count (CE – also referred to as the Gundersen coefficient), the point counting method option of the StereoInvestigator 9.0 software (MBF Biosciences) was applied. In all trials, the mean CE was 0.03.

To minimize confounds arising from surface irregularities, neuropil within a 2 µm 'guard zone' at the upper surface was not imaged. A neuron was counted if the top of its nucleus came into focus within the disector. If the nucleus was already in focus at the top of the 10 µm-thick optical disector the neuron was excluded (*Dodson et al., 2015*). As strongly fluorescent cytoplasmic/nuclear markers can obscure nuclear boundaries, delineation of these boundaries was achieved when necessary by incubating the tissue for 10 min in a solution (1:100,000) of the DNA dye 4',6-diamidino-2-

phenylindole (DAPI). For a given molecular marker, X, positive immunoreactivity (confirmed expression) is denoted as X+ while undetectable immunoreactivity (no expression) was denoted as X–. In all fluorescence analyses a neuron was classified as not expressing the tested marker only when positive immunoreactivity could be observed in other cells on the same optical section as the neuron in question. Each immunofluorescence protocol was repeated in a minimum of three adult rats/mice.

## Calculations of section area, volume and estimates of total cell number

Once the number of neurons expressing a marker or markers had been counted and recorded for a given coronal plane, the volume of the striatum in which these neurons had been counted (i.e. the optical disector) was first calculated. The cross-sectional area of a given delineated region was calculated using StereoInvestigator 9.0 software (MBF Biosciences). Since each set of counts was performed within the confines of an optical disector of a depth of 10 µm, the cross sectional area could be multiplied by this depth in order to obtain the volume of striatum in which the counts were performed for a given section. Because each count represents the total number of immunoreactive neurons (i.e. it is not an estimate) within this volume, dividing this number by the calculated volume yields the absolute density of immunoreactive neurons in that specific volume.

In order to calculate an estimate of the total number of neurons in the striatum, the following equation (*Oorschot, 1996*) was used:

$$N = Nv^* \, V(ref)$$

where N is the estimate for the total number of neurons in the structure, Nv is the numerical or volume density and V(ref) is the volume of that structure.

The volume of the dorsal striatum in rats and mice was calculated using the Cavalieri direct volume estimate (*Gundersen and Jensen, 1987*) with the following equation (*Oorschot, 1996*; *West, 1999*):

$$V(ref) = a(p)^* t \left( \sum P \right)$$

where P is the number of points counted within a delineated contour, a(p) is the area represented by each point and t is the fixed distance between each section. When calculated in this manner across the striatum of 9 rats, the mean volume of the dorsal striatum was determined to be 23.2 ± 2.1 $mm^3$. This value is similar to previous estimations for the volume of the dorsal striatum in the rat (*Oorschot, 1996*; *Rymar et al., 2004*). In a similar manner, the mean volume of the dorsal striatum of 3 mice was calculated to be 9.40 ± 0.29 $mm^3$. Once the value of V(ref) for the dorsal striatum in each animal is determined, multiplying that volume by the absolute neuronal density yields an estimate for the total number of neurons for that animal. Because most neuronal counts were performed across 3 animals (a sample that is typical of stereological studies), in depth statistics were deemed inappropriate, and so the value calculated from each animal is plotted along with the mean.

## Topographical and statistical analysis of interneuron distribution

In order to gain a quantitative understanding of any spatial biases in cell distributions, data obtained using StereoInvestigator regarding the position of individual interneurons and the volume of striatum imaged in a given coronal section were imported into MATLAB software (Mathworks, version: R2014b) for analyses with well-established Matlab toolboxes (see below). Due to the striatum's irregular shape across its 3 dimensions, horizontal and vertical lines were first 'drawn' from the position of each interneuron within a given coronal plane to the contours of striatum. Each interneuron was then assigned a value of between -1 and 1 according to its relative distance from the lateral and medial borders of striatum at that same coronal plane and to its distance from dorsal and ventral borders. A value of 0 indicated that an interneuron was positioned equidistantly from both striatal borders (e.g. the medial and lateral borders of striatum for the medio-lateral axis). Notably, all distance values were calculated according to the first points along the striatal contours that were contacted by the horizontal and vertical lines drawn from each interneurons position. In this manner, the definition of 'medial' and 'lateral' striatum was shifted depending on the position of the interneuron in the dorso-ventral plane; due to the irregular shape of the striatum. Likewise, the definition of 'dorsal' and 'ventral' striatum was also shifted depending on an interneuron's position in the medio-lateral plane.

As each counted interneuron in a section is given a value for each axis analyzed, and these values were subsequently pooled together, the sample size was large enough for the use of the Wilcoxon signed-rank test in order to determine if the mean relative value of a neural population in a given coronal plane differed significantly from 0, a value indicating an unbiased distribution along a given axis. In order to compare the mean positional values of two different interneuron populations along the medio-lateral or dorso-ventral axis within a given coronal plane, the Mann-Whitney U test was performed. When a section contained ≤5 neurons of a given type, positional analysis was not performed in either plane as variance became too great to give meaningful averages. Multiple comparisons were corrected using the false discovery rate (FDR) method in order to avoid false positives (*Noble, 2009*). The minimum significance level for all statistical tests was taken to be p≤0.05. To compare the variance in the distribution of neurons in the medio-lateral and dorso-ventral planes across the rostrocaudal axis, the difference in the mean normalised ML and DV values for a single coronal section and all other coronal sections was averaged. The absolute value of this measure could then be used to compare the variances of ML and DV positions of different interneuron types across different species using a Kruskal Wallis ANOVA. This measure was thus sensitive to changes in the mean position across the different coronal sections, but would yield low values if there was a consistent bias across all sections.

## Preparation of monkey tissue for indirect immunofluorescence

Monkey tissue used for stereological cell counting was obtained from two female adult rhesus monkeys (*Macaca mulatta*, 4.5–8.5kg) from the Yerkes National Primate Research Center colony. All procedures performed were in accordance with guidelines from the National Institute of Health and were approved by Emory's Animal Care and Use Committee. The highly experienced Yerkes veterinary staff took care of the animals. The Yerkes National Primate Research Center is an NIH-funded institution that is fully accredited by AAALAC, and regularly inspected by USDA. All activities at the Center are in compliance with federal guidelines. All experimental protocols concerning primates are performed in strict accordance with the NIH Guide for the Care and Use of Laboratory Animals and the PHS Policy on the Humane Care and Use of Laboratory Animals, and are reviewed and approved by the Emory IACUC before the proposed studies begin. Both monkeys were anaesthetized using an intravenous injection of pentobarbital (100 mg/kg). This was followed by a transcardial perfusion with cold oxygenated Ringer's solution, and 2 liters of a fixative solution containing 4% paraformaldehyde and 0.1% glutaraldehyde in phosphate buffer (0.1 M, pH 7.4). Each brain was then cut into 10 mm-thick blocks in the frontal plane, which were subsequently cut into 50 μm-thick sections using a vibrating-blade microtome. Sections containing the caudate/putamen were collected for use in immunofluorescence and cell counting procedures.

## Indirect immunofluorescence and stereological cell counting in Monkey Tissue

Seven coronal sections which extended from the 'head' to the beginning of the 'tail' of the caudate nucleus were selected from each brain for the purposes of indirect immunofluorescence and stereological cell counting. The seven sections from each monkey were matched such that the overall position of each section relative to Bregma from one monkey was similar to the sections chosen for the second monkey. The incubation procedure and immunofluorescence protocols (see *Table 1* for details) used to prepare the tissue are identical to those described for the rat and mouse. The delineation of the caudate and the putamen was defined using the Rhesus monkey brain atlas (*Paxinos et al., 2000*) and was done on a section by section basis. Once delineated, the acquisition, counting and positional analysis of immunolabeled neurons was carried out and represented in precisely the same manner as was done for the rodent (see above).

## In vivo electrophysiological recording and juxtacellular labeling of single neurons

Juxtacellular recording and labeling of neurons was performed in 137 anesthetized male Sprague-Dawley rats (280-350 g) according to previous protocols (*Sharott et al., 2012*) and in accordance with the Animals (Scientific Procedures) Act 1986 (UK). The induction of anesthesia was achieved using 4% v/v isoflurane in $O_2$, and maintained with urethane (1.3 g/kg, i.p; ethyl carbamate; Sigma),

and supplemental doses of ketamine (30 mg/kg, i.p.; Ketaset) and xylazine (3 mg/kg, i.p.; Rompun; Bayer). Wound margins were infiltrated with local anesthetic (0.5% w/v bupivacaine; Astra). A stereotaxic frame (Kopf) was used to fix the animal in place, and a homeothermic heating device (Harvard Apparatus) was used to maintain the animal's temperature at $37 \pm 0.5°C$. Recording of the electrocorticogram (ECoG) was performed directly above the frontal (somatic sensory-motor) cortex (4.0 mm rostral and 2.0 mm lateral of Bregma) (*Paxinos and Watson, 2007*)., with the reference being placed above the ipsilateral cerebellum (*Sharott et al., 2012*). Prior to acquisition, recordings of the ECoG subsequently underwent bandpass filtration (0.3–1500 Hz, −3dB limits) and amplification (2000X; DPA-2FS filter/amplifier; npi electronic). Using a computer-controlled stepper motor (IVM-1000; Scientifica), glass electrodes (10–30 MΩ in situ with a tip diameter of approximately 1.5 µm) filled with a 0.5 M solution of NaCl containing neurobiotin (1.5% w/v; Vector Laboratories) were lowered into the dorsal striatum under stereotaxic guidance. The depth of the electrode was constantly monitored and recorded at a resolution of 0.1 µm. Action potentials of striatal neurons (i.e. single-unit activity) were subsequently recorded after being amplified (Axoprobe-1A amplifier; Molecular Devices), AC-coupled, bandpass filtered at 300–5000 Hz (DPA-2FS filter/amplifier) and then amplified one more time by a factor of one hundred. Both the ECoG and single-unit activity were recorded at a sampling rate of 16.6 kHz using a Power1401 Analog-Digital converter and a PC running Spike2 acquisition and analysis software (Version 7.2; Cambridge Electronic Design).

Using the ECoG, the anaesthetized animal's brain state was defined and categorized as slow-wave activity (SWA) or cortical activation (*Magill et al., 2000*). Where possible, single unit activity was recorded during both of these states. Cortical SWA is characterized by the appearance of slow oscillations (approximately 1 Hz), which appear as alternating surface-positive 'active components', and surface negative 'inactive components' (*Mallet et al., 2008*) in the ECoG. Both spindle (7–12 Hz) and gamma (27–45 Hz) oscillations also appear in this brain state, which is analogous to that observed during drowsiness and non-rapid eye movement sleep (*Steriade, 2000*). In contrast, cortical slow and spindle oscillations are not present in the activated brain state, which is most similar to the brain activity recorded during the alert, behaving state.

Once a neuron was recorded, it was juxtacellularly labeled with the neurobiotin solution present in the glass electrode (*Sharott et al., 2012*). Labeling was achieved by sending continuous positive pulse currents (2–10 nA, 200 ms, 50% duty cycle) through the electrode until the recorded single-unit activity was 'entrained' by the injection of current. In most cases, this entrained state was maintained for 3 min or more in order to ensure adequate labeling. At the end of each experiment, animals were given a lethal dose of ketamine (150 mg/kg) followed by a transcardial perfusion using 50 ml of 0.05 M PBS, pH 7.4, followed by 300 ml of 0.1% w/v glutaraldehyde and 4% w/v paraformaldehyde (PFA) in 0.1 M PB, pH 7.4, and then by 200 ml of 4% w/v PFA in PB. Once retrieved, brains were left for a period of 12 hr in a fixative solution of 4% w/v PFA in PB. Each neuron was recorded and labelled in a different animal.

## Tissue processing for identification of recorded and labeled neurons

Using a vibrating microtome, 50 µm-thick parasagittal or coronal sections were serially collected and subsequently washed in PBS for 10 min. Sections were then incubated for a minimum of 4 hr in a solution of Triton PBS (PBS containing 0.3% v/v Triton X-100 [Sigma]) and Cy3-conjugated streptavidin (1:1000; Life Technologies). Sections containing neurobiotin-labelled cell bodies were isolated and subjected to further molecular characterization by indirect immunofluorescence. Neurons with dendritic spines were defined as spiny projections neurons (SPNs). Neurons that were suspected to be SPNs, but that did not have well labelled dendrites were tested for Ctip2 expression (*Arlotta et al., 2008*). Some confirmed SPNs were additionally tested for the expression of PPE (see below). Neurons that did not express the SPN marker Ctip2 and/or that possessed aspiny dendrites were tested for their expression of the major striatal interneuron markers, including parvalbumin (PV), choline acetyltransferase (ChAT), nitric oxide synthase (NOS) and neuropeptide Y (NPY). The initial molecular marker tested was guided by the labeled cell's somatodendritic structure, and once a positive expression for one of these markers was established, no other classical marker was tested since these molecules are rarely co-expressed, except for NOS and NPY (Kawaguchi et al., 1995). The expression of these markers was tested using the same indirect immunofluorescence protocol used for stereological cell counting.

After being washed one final time in PBS, sections were mounted on glass slides using Vecta-shield (Vector Laboratories) followed by the addition of a coverslip. Single-plane confocal images were acquired using a 1 µm optical disector and the 20X (Numerical Aperture = 0.8) or 40X (Numerical Aperture = 1.4) objective lens of a laser scanning confocal microscope (Zeiss LSM 710). For each section, absence of bleed through or cross talk between channels was investigated and was ensured by taking sequential and separate images. Importantly, neurons were declared 'negative' for a given marker if other neurons along the same focal plane were shown to express the marker under investigation. For neurons that were extensively filled with neurobiotin, multiple-plane images were taken in order to form a stack to highlight the somatodendritic structure. Linear brightness and contrast adjustments were made using Photoshop software (Creative Suite 3; Adobe Systems).

## Data selection and analysis of the firing of identified striatal interneurons

Periods of sustained, artifact free spontaneous SWA or spontaneous cortical activation were defined from the ECoG using previously described parameters (*Sharott et al., 2012*). The corresponding unit signal was digitally isolated using spike sorting procedures including principal component analysis (PCA) and template matching using Spike 2 software. Epochs with a minimum length of 60 s (range: 60–487 s) were converted into a binary digital signal that was imported into MATLAB for further analysis. The firing rate (spikes/s) of each recorded neuron was calculated as the number of spikes recorded within a selected data epoch. In order to perform comparisons of firing activity, non-parametric Kruskal–Wallis ANOVA on ranks, followed by Dunn's test for post hoc definition of significant pairwise differences were performed. The Mann-Whitney U or Wilcoxon signed rank test of significant pairwise differences were also used where appropriate. For all tests performed, the significance level was $p < 0.05$.

### Analysis of waveform length

In order to compare the waveforms of each identified PV+ interneuron, recorded spikes were divided into two segments, with the D1 segment representing the time taken for the waveform to go from baseline to its peak and the D2 segment representing the time taken for the waveform to return to the baseline from its peak. Recordings of identified striatal PV+/Scgn+ and PV+/Scgn− interneurons were used to calculate the mean length of the D1 and the D2 waveform segment for both interneuron types, as well as to calculate the length of the entire waveform. One identified PV+/Scgn+ interneuron was excluded from this analysis as its D2 segment could not be properly and consistently estimated. The lengths of the D1 and D2 segments as well as the total waveform lengths were compared using the Mann-Whitney U test. The minimum significance level for these statistical tests was taken to be $p \leq 0.05$

### Analysis of firing pattern

To determine the similarity of firing patterns between neurons, we calculated the Spearman-rho correlation coefficient between the ISI histograms of pairs of neurons. Thus neurons with very similar ISI histograms would have a correlation coefficient near 1, those that were unrelated near zero, and negatively correlated near -1. The correlation coefficients of specific pairs of neurons expressing particular combinations of molecular markers (i.e. PV+/Scgn+ or PV+/Scgn-) could then be compared statistically. This technique has previously been used to examine the similarity post-stimulation histograms of striatal neurons (*Adler et al., 2013*).

### Phase analysis

To investigate how the activity of individual striatal neurons varied in time with respect to ongoing cortical network activity, we analyzed the instantaneous phase relationships between striatal spike times and ongoing cortical oscillations. Electrocorticogram signals containing robust SWA or cortical activation were first filtered, using a neutral-phase bandpass filter (1–3 order Butterworth filter,) in 107 overlapping frequency bands with exponentially increasing center frequencies of 1-80 Hz and increasing widths of the plus/minus 1.5x the center frequency. The instantaneous phase and power of the ECoG in these frequency bands were separately calculated from the analytic signal obtained via the Hilbert transform (*Lachaux et al., 1999*). In this formalism, peaks in the ECoG oscillations

correspond to a phase of 0° and troughs to a phase of 180°. Circular phase plots and circular statistical measures were calculated using the instantaneous phase values for each spike in relation to each frequency band. Descriptive and inferential circular statistics were then calculated using the CircStat toolbox (*Berens, 2009*) for MATLAB. We included only those neurons that fired ≥40 spikes during the analyzed epoch. Phase histograms were constructed using 20 phase bins and used to construct pseudocolor plots of probability of firing as a function of phase for each frequency band. These neurons were then tested for significant phase-locked firing in each frequency band (defined as having p<0.05 in Rayleigh's Uniformity Test). The null hypothesis for Rayleigh's test was that the spike data were distributed in a uniform manner. The power of the ECoG in the SWA band was compared in data epochs used for analyses of phase-locked striatal neuron firing to test whether they were significantly different between groups. For frequencies below 5 Hz, Rayleigh tests were only carried out after any phase non-uniformities of the slow oscillations were corrected with the empirical cumulative distribution function (*Nakamura et al., 2014*; *Siapas et al., 2005*).

### iLFP peak histogram analysis

To investigate the timing of the activity of different striatal neuron populations in relation to local synchronized events, we calculated histograms of action potential firing times in relation to the slow oscillation (~1 Hz) present in the LFP recorded in striatum (using the same glass electrode that was used to simultaneously record single-unit activity). The polarity of the LFP was inverted (iLFP) so that peaks in the slow oscillation would correspond to the synchronized peaks in the membrane potential of many SPNs, often referred to as 'up states' (*Goto and O'Donnell, 2001*; *Wilson et al., 1990*). iLFP signals containing robust SWA were first filtered, using a neutral-phase bandpass filter (2nd order Butterworth filter) between 0.4 and 1.6 Hz. The instantaneous phase of the LFP slow oscillation the analytic signal obtained via the Hilbert transform. iLFP peaks were then defined as the period between each trough (i.e. –π to π). Each iLFP peak was then divided in to 100 time bins, irrespective of the total length of the peak in time, and a histogram of action potential firing was calculated for over these time bins (nominally assigned values from −1 to 1). Using this 'normalized time' approach, the time of action potential firing was not dependent on the duration of the iLFP peak, which can vary considerably between cycles. iLFP peaks that were shorter or longer than the period corresponding to the boundaries of the filter (i.e. 0.625 s and 2.5 s) were discarded. The median firing time of all spikes on the normalized time scale was used to compare the relationship of each neuronal population to the iLFP peak using a Kruskal-Wallis ANOVA on ranks with post-hoc Dunn tests.

### Hierarchical cluster analysis

Unsupervised clustering was performed to define the degree to which interneuron types could be segregated by their electrophysiological activity and the correlation between this functional segregation and their molecular identity. Clustering was performed using squared Euclidean distance in multidimensional space with Ward's linkage method (*Sosulina et al., 2010*). The number of clusters was established using a resampling variant of Thorndike's method, whereby the largest difference between successive within-cluster distances was used to determine the cutoff point. Clustering was performed on 10000 surrogate data sets created by randomly shuffling the value for each electrophysiological parameter and the differences between within cluster distances calculated for each surrogate. This null distribution, therefore, gave the within-cluster differences that would occur if the electrophysiological parameters were distributed randomly and was used to calculate p-values for the real data (i.e. the probability that the difference between successive within-cluster distances occurred by chance). The threshold for cluster separation was taken as the midpoint between the two within-cluster distances whose difference produced the lowest p-value < 0.05. Importantly for this study, using this criteria allowed for there to be a result where there were no significant clusters (i.e no p-value below 0.05).

Separate analyses were performed for SWA and cortical activation as interneuron rate, pattern and locking to cortical oscillations vary considerably between these brain states (*Sharott et al, 2012*). For SWA the following parameters were included in the analysis: the log of the 10th percentile of interspike interval (Log. ISI 10), the ratio of the proportion CV2 values lower than 0.2 and higher than 1.85 (CV2 ratio), the median firing time of spikes in relation normalized iLFP

peak (iLFP peak), the vector length of the phases of all action potentials in relation to the ECoG slow (0.4–1.6 Hz), spindle (7–12 Hz) and gamma (30-48 Hz) oscillations (ECoG SWA vec, ECoG spin. vec, ECoG gam vec., respectively). For cortical activation, Log. ISI 10 and CV2 ratio were also used in addition to the log of the 50th percentile of interspike interval (Log. ISI 50), the log of the 85th percentile of interspike interval (Log. ISI 85), the mean of the CV2 (CV2 mean) and the mean firing rate (Rate). These parameters were chosen based on their ability to separate previously established categories of striatal interneuron with well-defined molecular, structural, and electrophysiological features, (cholinergic/choline acetyltransferase-expressing (ChAT+) interneurons, GABAergic nitric oxide synthase /neuropeptide Y-expressing (NOS+/NPY+) interneurons and PV+ interneurons, irrespective of Scgn-expression), and were based on previous analyses of the firing of these of these cell types in vivo (*Sharott et al, 2012*). This approach allowed us to evaluate the clustering of PV+/Scgn+ against well-established interneuron types using parameters that were generally effective for classifying interneurons, rather than being selected to discriminate specific populations.

## Identification and quantification of axonal boutons of juxacellularly-labeled interneurons

After in vivo recordings and effective juxtacellular labeling of select interneurons, all sections containing intense Cy3 signal for neurobiotin were isolated. The section(s) containing the cell body were separated from those containing neurobiotin-filled dendrites or axons. These cell body containing sections were then subject to tests for PV and Scgn immunohistochemistry in order to determine whether the labeled cell expressed one, both or neither of these proteins. If a cell was determined to be PV+/Scgn+ or PV+/Scgn-, all sections containing the neurobiotin filled axons of the corresponding cell were incubated with the following primary antibodies (see *Table 1* for details): goat anti-DARPP-32, rabbit anti-preproenkephalin (PPE) and mouse anti-gephyrin using the same 2-step incubation method described above for stereological cell counting. As simultaneous detection of these 3 markers (in addition to the detection of the neurobiotin signal) was not possible due to technical limitations, some axon containing sections were incubated with antibodies against DARPP-32 and gephyrin while others were incubated with antibodies against DARPP-32 and PPE. The presence of a DARPP-32 signal was used to mark the somata of spiny projection neurons. Gephyrin, a post-synaptic protein involved in the anchoring of inhibitory receptors to the neuron's cytoskeleton (*Charrier et al., 2006*) was used as a marker of putative GABAergic synapses. For protocols involving DARPP-32 and gephyrin detection of gephyrin expression at the point of contact between a neurobiotin-filled axon bouton and the edge of the DARPP-32 signal was used as indirect evidence of the existence of synapses between axonal boutons and SPN somata. For protocols involving DARPP-32 and PPE, the expression of PPE was used to determine whether a DARPP-32 expressing SPN was of the direct pathway (PPE-) or indirect pathway (PPE+).

After one final wash in PBS, each section was mounted on a glass slide in Vectashield and multiple single-plane confocal images (or stacks) were acquired using a 1 μm optical disector and a 63X objective lens (Numerical Aperture = 1.4) of a laser scanning microscope (Zeiss LSM 710). Appropriate sets of laser beams and emission windows were used for Alexa Fluor 488 and DyLight 488 (excitation 488 nm, emission 492–544 nm), Cy3 (excitation 543 nm, emission 552–639 nm), and Cy5 and DyLight649 (excitation 633 nm, emission 639–757 nm).

## Analysis and quantification of appositions made by the axons of identified interneurons

Captured images of axon containing sections were analyzed using Zeiss imaging software (Carl Zeiss Microscopy, GmbH). Sections incubated with antibodies against DARPP-32 and gephyrin were analyzed in order to establish how often the swelling of an axonal bouton apposed to a DARPP-32 expressing SPN soma was co-localized with a positive gephyrin signal, signifying that these appositions likely indicated the presence of putative synapses. Other sections containing neurobiotin-filled axons were incubated with antibodies against DARPP-32 and PPE. In such sections, putative synapses made between the axons of labelled interneurons and SPNs could not be identified using gephyrin, and appositions were instead defined when an axonal bouton (a swelling of the axon) was in close proximity (~0.1 μm) to the cell body of a DARPP-32+ SPN. These appositions could be

viewed in 3 planes using the software to ensure that the swelling of the axon defined as a bouton was consistent in 3 dimensions and was not the result of overexposure of the neurobiotin signal. Appositions were established after verification that there were no detectable pixels between the cell body signal and the neurobiotin signal of the axonal bouton. For each section, the number of boutons on the neurobiotin-filled axon of an identified striatal interneuron was counted.

These counts included those boutons that apposed the cell bodies of SPNs as well as those that did not. In cases where a bouton contacted an SPN cell body, the expression of PPE (or lack thereof) was used to determine whether that SPN was of the direct pathway (PPE– SPN) or the indirect pathway (PPE+ SPN). The number of appositions with PPE– and PPE+ somata was counted along with the total number of PPE+ and PPE– SPNs that were positioned within a 30 μm radius from any neurobiotin filled axons.

In total, the neurobiotin-filled axons of 4 PV+/Scgn- and 4 PV+/Scgn+ interneurons were assessed in this manner, which allowed for the counting of 100 s of boutons. This encompassed all juxtacellularly-labelled neurons with well-labelled axon on 3 or more sections and any section from these neurons with ≥10 axonal swellings was included in 1 or more protocols. For each neuron, boutons were classified as either apposed to SPN cell bodies or not apposed. Those boutons that were apposed were further classified as apposed to PPE+ or PPE- cell bodies. The counts for each neuron were averaged with those of other neurons belonging to the same group (i.e. PV+/Scgn- or PV+/Scgn+). Statistical comparisons comparing whether one group significantly differed from another were performed using the Fisher exact test. The minimum significance level for these statistical tests was taken to be $p \leq 0.05$.

## Acknowledgements

This work was supported by the Medical Research Council UK (MRC; awards MC_UU_12020/5 and MC_UU_12024/2 to PJM, and award MC_UU_12024/1 to AS), a Marie Curie European Re-integration Grant (SNAP-PD) awarded by the European Union, and the Wellcome Trust (Investigator Award 101821 to PJM). FNG and EK were supported by the MRC and University of Oxford Clarendon Fund Scholarships. FV was supported by an MRC studentship. RS was in receipt of a Wellcome Trust Clinical Fellowship (WT_RS_109030/Z/15/Z). We thank T Harkany and R Faram for early insights on secretagogin, and G Hazell, B Micklem, L Norman, K Whitworth, J Janson, L Conyers and C Johnson for technical support. We also thank G. Silberberg and K. Meletis for their expert insights.

## Additional information

### Funding

| Funder | Grant reference number | Author |
|---|---|---|
| University Of Oxford | Clarendon Fund Scholarship | Farid N Garas<br>Eszter Kormann |
| Medical Research Council | | Farid N Garas<br>Eszter Kormann |
| Wellcome Trust | 109030/Z/15/Z | Rahul S Shah |
| Medical Research Council | 1531859 | Eszter Kormann |
| Medical Research Council | Studentship | Federica Vinciati |
| Human Frontier Science Program | LT000396/2009-L | Kouichi C Nakamura |
| Medical Research Council | MC_UU_12020/5 | Peter J Magill |
| Medical Research Council | MC_UU_12024/2 | Peter J Magill |
| Wellcome Trust | 101821 | Peter J Magill |
| European Research Council | SNAP-PD | Peter J Magill<br>Andrew Sharott |
| Medical Research Council | MC_UU_12024/1 | Andrew Sharott |

| National Institutes of Health | NIH base grant to the Yerkes National Primate Research Center P51 OD011132 | Yoland Smith |

The funders had no role in study design, data collection and interpretation, or the decision to submit the work for publication.

## Author contributions

FNG, AS, Conception and design, Acquisition of data, Analysis and interpretation of data, Drafting or revising the article; RSS, EK, NMD, YS, Acquisition of data, Drafting or revising the article; FV, KCN, Conception and design, Acquisition of data; MCD, Conception and design, Contributed unpublished essential data or reagents; PJM, Conception and design, Analysis and interpretation of data, Drafting or revising the article

## Author ORCIDs

Rahul S Shah, http://orcid.org/0000-0002-8616-8693
Natalie M Doig, http://orcid.org/0000-0001-6821-0264
Kouichi C Nakamura, http://orcid.org/0000-0001-7053-0507
Matthijs C Dorst, http://orcid.org/0000-0002-1162-1481
Peter J Magill, http://orcid.org/0000-0001-7141-7071
Andrew Sharott, http://orcid.org/0000-0002-1594-3349

## Ethics

Animal experimentation: All rodent experiments in this study were performed in strict accordance with the Animals (Scientific Procedures) Act, 1986 (UK) under Home Office project licences 30/3159 and 30/2629. All non-human primate experiments in this study were performed in strict accordance with guidelines from the National Institute of Health and were approved by Emory's Animal Care and Use Committee. All procedures for the primate work were performed in accordance with guidelines from the National Institute of Health and were approved by Emory's Animal Care and Use Committee. The highly experienced Yerkes veterinary staff took care of the animals. The Yerkes National Primate Research Center is an NIH-funded institution that is fully accredited by AAALAC, and regularly inspected by USDA. All activities at the Center are in compliance with federal guidelines. All experimental protocols concerning primates are performed in strict accordance with the NIH Guide for the Care and Use of Laboratory Animals and the PHS Policy on the Humane Care and Use of Laboratory Animals, and are reviewed and approved by the Emory IACUC before the proposed studies begin.

## Additional files

### Supplementary files

• Supplementary file 1. Distribution of PV+/Scgn– and PV+/Scgn+ interneurons in the rat striatum.

• Supplementary file 2. Distribution of PV+/Scgn– and PV+/Scgn+ interneurons in the macaque caudate and putamen.

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
