## [Decision Letter]

Thank you for submitting your article "Secretagogin expression delineates functionally-specialized populations of striatal parvalbumin-containing interneurons" for consideration by *eLife*. Your article has been reviewed by three peer reviewers, and the evaluation has been overseen by a Reviewing Editor and Gary Westbrook as the Senior Editor. The following individuals involved in review of your submission have agreed to reveal their identity: Tibor Koos (Reviewer #2); Jochen Roeper (Reviewer #3). The reviewers have discussed the reviews with one another and the Reviewing Editor has drafted this decision to help you prepare a revised submission.

Summary:

This study investigates the properties of subpopulations of PV+ interneurons in the striatum identified on the basis of co-expression of Secretagogin (Scgn) using anatomical and neurophysiological methods. The results demonstrate convincing anatomical and electrophysiological differences between the two neuron populations confirming that the differential expression of Scgn reveals the existence of functionally distinct cell types. However, there are a few concerns and further analyses that need to be considered.

1) The reviewers agree that there are distinct populations of PV interneurons in striatum, but there are concerns about using Scgn as the main marker for this division. The reviewers request evidence of the segregation into clusters using unsupervised clustering of the all the data but leaving out Scgn expression. This would significantly increase the confidence in the results. It should also be then verified if these population divisions are not seen in mice, independently of Scgn expression (or if it is only the expression of Scgn that does not happen).

2) The reviewers suggest that the main claims of connectivity be checked electrophysiologically, namely a) the differential innervation of putative direct versus indirect pathway medium spiny neurons; and b) the claim that the PV/Scgn positive striatal interneurons target preferentially the dendrites rather than the soma.

The reviewers assess that these data could be collected/analyzed in a timely manner.

Below are their individual comments for your benefit.

Reviewer #1:

The manuscript by Garas et al., describes a population of PV/Scgn positive striatal interneuron in the rat and primate but not in the mice. They describe a differential anatomical distribution and proportion of this newly described striatal populations when comparing it to the PV positive /Scgn negative striatal interneurons in the rat. Interestingly they report a bigger proportion of the PV/Scgn positive striatal interneuron in the primate caudate-putamen. Furthermore by performing electrophysiological recordings in anesthetized rats they show that the spiking of the PV/Scgn positive vs. the PV positive /Scgn negative striatal interneuron is differently locked to cortical oscillations. Finally by performing elegant immunofluorescence experiments from some of the recorded/labeled cells in vivo they identify that the PV/Scgn positive striatal interneurons targeted preferentially the direct pathway SPN. Based on the analysis of the presented results, the fact that this newly described population of striatal interneurons is described in the absence of genetically modified manipulations in primates and rodents, and that the results fulfill the current requirements to describe a population of neurons (anatomical marker and functional specificities) make this reviewer incline to accept this manuscript for publication in *eLife* considering the following comments and when stated further experiments.

1) The anatomical experiments presented in this manuscript are very convincing however a functional demonstration of the specific targeting of the newly described population of striatal interneurons on the direct pathway SPN is needed, otherwise this argument must be presented as a suggestion rather than a conclusion.

2) As in point 1 the argument to claim that the newly described PV/Scgn positive striatal interneurons target preferentially the dendrites rather than the soma must be addressed with functional experiments otherwise must be presented as suggestion rather than a conclusion.

3) There is a recently published paper by Muñoz-Manchado et al., (PMID: 25146369) describing that the PV striatal interneurons are composed of more than one single population of interneurons in mice (based on the expression of the 5HT receptor). The findings presented in that paper must be place into consideration when discussing here the PV/Scgn positive striatal cells as a new population of striatal interneurons and further discussion about which will be the requirements in the future to define a population of striatal interneurons may be included.

4) The description of the new population of PV/Scgn positive striatal cells in rats/primates but not in mice, raised the following: Does the absence of the PV/Scgn positive striatal interneuron in mice remain if using animals of different ages? Scgn has been used as marker for migrating cells. I noted that in the experiment presented rats were older than mice…

5) May a more complete anatomical comparison of the axonal and dendritic arborization be presented to judge whether the two different populations may impact on different quantities of SPN.

Reviewer #2:

Summary:

The study investigates the properties of subpopulations of PV+ interneurons identified on the basis of co-expression of Secretagogin (Scgn) using anatomical and neurophysiological methods. The results demonstrate convincing anatomical and electrophysiological differences between the two neuron populations confirming that the differential expression of Scgn reveals the existence of functionally distinct cell types.

Critique:

This is an outstanding and highly interesting study, providing the first demonstration of the existence of a functionally significant division within the population of PV expressing striatal interneurons in the neostriatum. The demonstration of distinct PV+ neurons may help to clarify outstanding issues about the role of previously considered FSIs both in vitro and in vivo. The data are very high quality and the large number of observations provide extra confidence in the conclusions.

I have only minor points of criticism:

1) Since 1/4 of PV+ cells are Scgn+ and most of them are caudal, and further, as the Scgn- PV neurons are evenly distributed in the medial-lateral directions the observations seem to be at odds with the previously reported difference in PV cell density between the medial and lateral striatum. The authors should discuss why their results are different from previous observations (e.g.: Luk and Sadikot 2001; etc.).

2) It would be of great interest if the authors could comment on the morphological difference between the 2 types of PV neurons, in particular with respect to relative extent of axonal their arborization. Do these cell types correspond to the previously suggested local and extended FSIs (Kawaguchi 1993)?

Reviewer #3:

A small set of calcium-binding proteins (CBPs e.g. calbindin, calretinin & parvalbumin) has been traditionally used to label and categorize neuronal subpopulation – among them principal cells as well as interneurons. Facilitated by comparatively high expression levels and excellent commercially available antibodies, these makers are still very useful and popular – however mainly as heuristic tools. For advanced cell-type classification this small set of markers are not state-of-the-art anymore and have been succeeded by more comprehensive neuronal classifications based on in-depth and combined analysis of functional properties, connectivity and – importantly – co-expression of much larger gene set up to full single-cell transcriptomes.

As has been noted before (Mulder et al. 2009, PNAS) – sectretagogin (SCGN) is differentially co-expressed in certain neuronal populations across different species (mouse, rat, primate). However, with its functional significance remaining obscure, this fact remains little more than an oddity that is not unique to striatal interneurons as reported by Garas and colleagues. The differential regional distribution of SCGN+/PV+ and SCGN-/PV+ interneurons across the caudal-rostral axis of the striatum is interesting but – in the current state of the study – again does not lead to further insights or testable hypothesis. Thus, like many other results of this study – all technically sound and interesting findings in their own right – they are interesting starting points in need of further more focused exploration. The electrophysiological characterization of the SCGN+/PV+ and SCGN-/PV+ neurons presented by the authors and the respective coupling to cortical activity states does show partial overlap and argues rather for gradual differences between striatal interneurons than presenting a strong case for categorical differences – as for comparison is well established for distinct PV+ subpopulations in hippocampus. Also, the differential innervation of putative direct versus indirect pathway medium spiny neurons is another interesting starting point of an in depth study evaluation the functional consequences of this finding. In summary, this study is a – relatively loose – collection of interesting findings – however, none of them is really followed up to reveal important novel insights about distinct functional classes of striatal interneurons – expected to be delivered to warrant publication in a top journal and of interest to wide readership. As it stands, it is a useful data collection for experts interested in novel tools to segregate striatal interneurons.

---

## [Author Response]

1) The reviewers agree that there are distinct populations of PV interneurons in striatum, but there are concerns about using Scgn as the main marker for this division. The reviewers request evidence of the segregation into clusters using unsupervised clustering of the all the data but leaving out Scgn expression. This would significantly increase the confidence in the results. It should also be then verified if these population divisions are not seen in mice, independently of Scgn expression (or if it is only the expression of Scgn that does not happen).

We agree that unsupervised cluster analysis can be a useful additional technique for distinguishing populations of neurons. It should be noted, however, that not all data sets are amenable to such cluster analysis. We have now performed an unsupervised hierarchical cluster analysis of the electrophysiological parameters of interneurons we recorded in vivo (see below). Although we were not able to perform a cluster analysis of our anatomical (cell count) data, we have nevertheless performed a new complementary analysis to address the spirit of the reviewer’s comment here. More specifically, we have now provided careful estimates of whether and to what extent the spatial distributions of all PV+ interneurons as a whole could be explained by the selective expression of Scgn (see Results subsection “Selective expression of secretagogin divides the PV+ interneuron population in rat striatum but not in mouse striatum”, last two paragraphs and new Figure 2). We now show in rats that the distribution biases of all PV+ interneurons are similar to those of PV+/Scgn+ interneurons, suggesting the presence of PV+/Scgn+ interneurons biases the entire PV+ interneuron population. In line with this, there was no consistent or strong bias in the relative positions of PV+/Scgn- interneurons. To further explore the notion that the biased distributions of rat PV+ interneurons can be largely explained by Scgn expression, we also analyzed the spatial distributions of all PV+ interneurons in the mouse dorsal striatum, which contains a tiny number of PV+/Scgn+ interneurons. There was no consistent or strong bias in the relative positions of all PV+ interneurons in mice. Thus, we can now conclude from this new analysis, that: (1) the highly unusual spatial distribution of rat PV+/Scgn+ interneurons accounts for much of the non-uniform distribution of all PV+ interneurons in rats; and (2) mouse PV+ interneurons lack Scgn and are more uniformly distributed. Taken together, these data serve to reinforce that Scgn is a useful and highly relevant marker for dividing PV+ interneuron populations.

In our original submission, we provided evidence of PV+/Scgn- and PV+/Scgn+ interneurons differing in several of their electrophysiological properties, suggesting they are discrete cell types. To increase confidence in these initial results, we have now performed an unsupervised hierarchical cluster analyses of the electrophysiological parameters of PV+ interneurons (omitting information as to whether or not they expressed Scgn) together with those of cholinergic interneurons and GABAergic interneurons that co-express nitric oxide synthase (NOS) and neuropeptide Y (NPY). We included ChAT+ interneurons and NOS+/NPY+ interneurons because they are widely accepted to be discrete cell types and, as such, can be used as comparators; if PV+/Scgn- and PV+/Scgn+ interneurons can be distinguished to the same degree as ChAT+ and NOS+/NPY+ interneurons can be distinguished (from each other and/or from the subpopulations of PV+ interneurons), then this would support the notion that PV+/Scgn- and PV+/Scgn+ interneurons are discrete cell types. Another key advantage of including ChAT+ and NOS+/NPY+ interneurons is that they facilitated the unbiased selection of the electrophysiological parameters to analyze; parameters were thus selected according to their utility for segregating one or more of the populations of ChAT+ interneurons, PV+ interneurons (as a whole) and NOS+/NPY+ interneurons, rather than their ability to distinguish PV+/Scgn- from PV+/Scgn+ interneurons per se. The cluster analysis was performed with standard methods (Squared Euclidian Distance, Ward’s method) together with non-parametric resampling-based version of Thorndike's method to provide a probabilistic evaluation of the number of clusters (which, importantly, using this method could be 1. i.e. no segregation). As now shown in new Figure 10 and reported in the Results (subsection “Hierarchical clustering of striatal interneurons using electrophysiological parameters is highly correlated with their molecular identities”), the cluster analysis showed that interneurons segregated into significant clusters. After assignment of molecular identities, it was evident that these clusters were predominantly composed of ChAT+ interneurons, NOS+/NPY+ interneurons, PV+/Scgn+ interneurons, and PV+/Scgn- interneurons, respectively. The cluster analysis also showed that PV+/Scgn+ and PV+/Scgn- interneurons are as distinct from each other as either subpopulation is from NOS+/NPY+ or ChAT+ interneurons. In brief, these unsupervised cluster analyses confirm not only that PV+ interneurons can be segregated into two discrete groups on the basis of their electrophysiological properties alone (in two brain states), but also that these discrete interneuron groups differ with respect to their Scgn expression. Taken together, these data further support the notion that rat PV+ interneurons are comprised of two main subpopulations, and that selective expression of Scgn is a robust and useful metric for distinguishing between them.

We would like to emphasize for the reviewers that the recording and juxtacellular labelling of single interneurons in striatum in vivo is extremely challenging, not least because interneurons make up such a small proportion of all striatal cells. Indeed, the rat interneuron data we present in this paper required 100+ experiments carried out over 5+ years. Thus, it is reasonable to state that electrophysiological recordings from identified PV+/Scgn- and PV+/Scgn+ interneurons in mice are well beyond the scope of this study (and the journal’s timeline for re-submission). This being the case, we cannot explore a population division in mice beyond our careful quantification of Scgn expression. The results of our anatomical analysis are unambiguous; Scgn expression is very rare in mouse PV+ interneurons. However, we have taken care to state in the Discussion (third paragraph) that it is still possible that mouse striatum contains a sub-population of PV+ interneurons that have the same anatomical and physiological properties as the PV+/Scgn+ interneurons in rats.

*2) The reviewers suggest that the main claims of connectivity be checked electrophysiologically, namely a) the differential innervation of putative direct versus indirect pathway medium spiny neurons; and b) the claim that the PV/Scgn positive striatal interneurons target preferentially the dendrites rather than the soma.*

Thank you for these interesting suggestions. We foresee that one might adopt in vitro and/or in vivo strategies to provide electrophysiological correlates of the selective innervation of iSPNs and dSPNs.

In terms of an in vitro strategy, intracellular recordings of synaptically-connected pairs of PV+ interneurons and SPNs in brain slices could provide important insights. However, with respect to our study, which is necessarily focused on the use of wildtype adult animals, there are several major obstacles to be overcome in such experiments: (1) PV+/Scgn+ and PV+/Scgn- neurons in living slices cannot be visualised in advance to guide the recordings. This means that less than 1 in 100 intracellular recordings would yield an interneuron of the desired type. It would take much time to gather a sufficiently robust sample of interneurons; (2) There is a confounding combinatorial factor, in that 4 types of connection would have to be studied before firm conclusions could be drawn, i.e. PV+/Scgn- to dSPN, PV+/Scgn- to iSPN, PV+/Scgn+ to dSPN and PV+/Scgn+ to iSPN; (3) The synaptic connections in question are selective, not specific. This means that statistical verification of preferential connectivity would require many pairs exhibiting each of the 4 types of connection. (4) Synaptic connectivity can be sparse in slices. The relevant literature suggests that synaptic connectivity will not be detected in a substantial number of paired recordings; (5) The molecular identities of all record neurons would have to be verified post hoc; and (6) All recordings would have be made in slices taken from adult animals. In summary then, it is clear that such in vitro work is far beyond the technical and temporal scope of the current study. However, we have now highlighted in the Discussion (eighth paragraph) that, in the future, in vitro recordings of synaptically-connected pairs of neurons should provide useful insights.

To address the reviewers’ comment, we have thus adopted an in vivo strategy that does not necessitate paired or intracellular recordings of neurons. Our new experiments and analyses are not only designed to provide electrophysiological correlates of the selective innervation of iSPNs and dSPNs, but also to simultaneously address an influential concept, namely that striatal PV+ interneurons are considered to provide “feedforward” inhibition to SPNs. Our anatomical data suggest that PV+/Scgn- and PV+/Scgn+ interneurons selectively innervate iSPNs and dSPNs, respectively. When our anatomical observations are placed within a framework of feedforward inhibition in vivo, one might predict that, first, the firing of PV+/Scgn- interneurons is more likely to precede the firing of iSPNs than dSPNs, and secondly, that the firing of PV+/Scgn+ interneurons is more likely to precede the firing of dSPNs than iSPNs. To test these predictions, we have recorded robust samples of identified dSPNs and iSPNs in anesthetized rats during cortical SWA (i.e. under conditions identical to those under which the PV+ interneurons were recorded). We then quantitatively compared the spike timings of these SPNs with those of the PV+/Scgn- and PV+/Scgn+ interneurons. As now shown in new Figure 9 and reported in the Results (subsection “PV+/Scgn- and PV+/Scgn+ interneurons differ in their temporal relationships with striatal projection neurons of the direct and indirect pathways”), we have discovered that: (1) the spike timings of dSPNs and iSPNs are distinct; (2) the spike timings of PV+/Scgn- and PV+/Scgn+ interneurons are distinct; (3) PV+/Scgn- interneurons fire significantly earlier than iSPNs but not dSPNs; and (4) PV+/Scgn+ interneurons do not fire before iSPNs or dSPNs. These electrophysiological data verify the first prediction above with respect to possible substrates for feedforward inhibition, that is, the firing of PV+/Scgn- interneurons is indeed more likely to precede the firing of iSPNs than dSPNs. This result is consistent with our anatomical data. The firing of PV+/Scgn+ interneuron firing with respect to dSPN firing is not consistent with a powerful feedforward inhibitory connection to either type of SPN. Again, this result resonates with our anatomical data showing that, although PV+/Scgn+ interneuron preferentially target dSPNs, these interneurons form fewer putative somatic synapses and fewer appositions on a given SPN soma. Taken together, these new experiments provide a compelling corroboration of our anatomical data, and further reinforce our original conclusion that the diverse properties of PV+/Scgn- and PV+/Scgn+ interneurons position them to fulfil different roles in the striatal circuit.

In the original submission, we were careful not to claim any preferential innervation of the dendrites of any striatal cell type. Nevertheless, in the Discussion (sixth paragraph), we have now made it clear that we can only speculate on the identities of other targets innervated by PV+/Scgn+ and PV+/Scgn- interneurons.

The reviewers assess that these data could be collected/analyzed in a timely manner.

Below are their individual comments for your benefit.

Reviewer #1:

1) The anatomical experiments presented in this manuscript are very convincing however a functional demonstration of the specific targeting of the newly described population of striatal interneurons on the direct pathway SPN is needed, otherwise this argument must be presented as a suggestion rather than a conclusion.

As described above, we have added a new data set (based on in vivo recordings of identified SPNs) that support some of our predictions of the functional impact of selective targeting of SPNs by PV+/Scgn- and PV/Scgn+ interneurons. Nevertheless, we have toned down our statements in line with the reviewer’s comment here: see Abstract and Introduction (last paragraph). We have also highlighted in the Discussion (eighth paragraph) that, in the future, in vitro recordings of synaptically-connected pairs of neurons should provide useful insights into the predicted functional impact of the selective targeting of dSPNs and iSPNs by different sets of PV+ interneurons.

*2) As in point 1 the argument to claim that the newly described PV/Scgn positive striatal interneurons target preferentially the dendrites rather than the soma must be addressed with functional experiments otherwise must be presented as suggestion rather than a conclusion.*

In the original submission, we were careful not to claim any preferential innervation of the dendrites of any striatal cell type. Nevertheless, we appreciate the reviewer’s point here and, in the Discussion (sixth paragraph), we have now made it clear that we can only speculate on the identities of other targets innervated by PV+/Scgn+ and PV+/Scgn- interneurons.

3) There is a recently published paper by Muñoz-Manchado et al., (PMID: 25146369) describing that the PV striatal interneurons are composed of more than one single population of interneurons in mice (based on the expression of the 5HT receptor). The findings presented in that paper must be place into consideration when discussing here the PV/Scgn positive striatal cells as a new population of striatal interneurons and further discussion about which will be the requirements in the future to define a population of striatal interneurons may be included.

We thank the reviewer for reminding us of the relevance of this excellent study. We have now cited this paper. In the Introduction (second paragraph), we now highlight some of its key findings pertaining to the molecular heterogeneity of PV+ interneurons. In the Discussion (third paragraph), we now put our findings in context of interneuron heterogeneity defined by selective expression of 5HT receptor subunits, and highlight some lines of investigation for the future.

4) The description of the new population of PV/Scgn positive striatal cells in rats/primates but not in mice, raised the following: Does the absence of the PV/Scgn positive striatal interneuron in mice remain if using animals of different ages? Scgn has been used as marker for migrating cells. I noted that in the experiment presented rats were older than mice.

In our original submission, we stated the age of the mice as 6 weeks. This was incorrect, please accept our apologies. The adult mice used in this study were 3-4 months old. We have now corrected this error in the Methods section (first paragraph). Importantly then, the ages of the mice and rats were equivalent, suggesting that the lack of Scgn expression in mouse striatum cannot be attributed to any potential differences in ‘developmental stage’. The focus of our paper is on adult animals; quantifying Scgn expression across the life course of multiple species is beyond the scope of this work.

5) May a more complete anatomical comparison of the axonal and dendritic arborization be presented to judge whether the two different populations may impact on different quantities of SPN.

We appreciate the reviewer’s comment here. However, we would like to emphasise that accurate reconstruction of the axons and dendrites of several PV+/Scgn+ interneurons as well as several PV+/Scgn- interneurons is needed to address this point. Drawing on our experience of cell reconstructions, and having consulted with experts in the field (Paul Bolam, Gilad Silberberg) who have attempted to reconstruct the axons of striatal PV+ interneurons, we conservatively estimate that each interneuron reconstruction would take several weeks of full-time work to complete. In short, the required constructions cannot be delivered in a timely manner and that would be compliant with the journal’s request to re-submit our paper within two months.

Reviewer #2:

I have only minor points of criticism:

1) Since 1/4 of PV+ cells are Scgn+ and most of them are caudal, and further, as the Scgn- PV neurons are evenly distributed in the medial-lateral directions the observations seem to be at odds with the previously reported difference in PV cell density between the medial and lateral striatum. The authors should discuss why their results are different from previous observations (e.g.: Luk and Sadikot 2001; etc.).

We appreciate the reviewer’s point here: According to Luk & Sadikot (2001), the density of PV+ interneurons in rat “dorsolateral striatum” is higher than that in “ventromedial striatum”. To address the reviewer’s comment, we have performed an additional analysis of the distributions of all PV+ interneurons irrespective of Scgn expression (as Luk & Sadikot would have counted). The results of our new analysis are presented in new Figure 2 and discussed in the Results subsection “Selective expression of secretagogin divides the PV+ interneuron population in rat striatum but not in mouse striatum”, last two paragraphs. In short, we found no quantitative evidence of a consistent or clear “dorsolateral” bias in the distribution of all PV+ interneurons in the striatum of rats. This discrepancy with respect to Luk & Sadikot (2001) presumably arises from differences in the extent to which different regions of striatum were sampled as well as in the cell-counting methodologies/analyses used in each study. For example, we sampled from 13 coronal planes (rather than 1 plane as used by Luk & Sadikot) and we calculated the relative position of each counted neuron in the mediolateral or dorsoventral axes (rather than calculate densities, which would require binning of counts into arbitrarily defined areas, such as the “quadrants” used by Luk & Sadikot). Our method of quantifying biases in spatial distributions is comprehensive and objective, thus offering several advantages as compared to previous methods.

2) It would be of great interest if the authors could comment on the morphological difference between the 2 types of PV neurons, in particular with respect to relative extent of axonal their arborization. Do these cell types correspond to the previously suggested local and extended FSIs (Kawaguchi 1993)?

It would be premature for us to comment further on the structural properties of the interneurons we have recorded and labelled in vivo. Accurate reconstruction of the dendrites and axons of several PV+/Scgn+ interneurons as well as several PV+/Scgn- interneurons is needed to address this point (please see above for explanation why z-stacks are challenging and could be misleading). Drawing on our experience of cell reconstructions, and having consulted with experts in the field (Paul Bolam, Gilad Silberberg) who have attempted to reconstruct the axons of striatal PV+ interneurons, we conservatively estimate that each interneuron reconstruction would take several weeks of full-time work to complete. Given the journal’s request to re-submit our paper within two months, the required constructions cannot be delivered in a timely manner.

Reviewer #3:

A small set of calcium-binding proteins (CBPs e.g. calbindin, calretinin & parvalbumin) has been traditionally used to label and categorize neuronal subpopulation – among them principal cells as well as interneurons. Facilitated by comparatively high expression levels and excellent commercially available antibodies, these makers are still very useful and popular – however mainly as heuristic tools. For advanced cell-type classification this small set of markers are not state-of-the-art anymore and have been succeeded by more comprehensive neuronal classifications based on in-depth and combined analysis of functional properties, connectivity and – importantly – co-expression of much larger gene set up to full single-cell transcriptomes.

As has been noted before (Mulder et al. 2009, PNAS) – sectretagogin (SCGN) is differentially co-expressed in certain neuronal populations across different species (mouse, rat, primate). However, with its functional significance remaining obscure, this fact remains little more than an oddity that is not unique to striatal interneurons as reported by Garas and colleagues. The differential regional distribution of SCGN+/PV+ and SCGN-/PV+ interneurons across the caudal-rostral axis of the striatum is interesting but – in the current state of the study – again does not lead to further insights or testable hypothesis. Thus, like many other results of this study – all technically sound and interesting findings in their own right – they are interesting starting points in need of further more focused exploration. The electrophysiological characterization of the SCGN+/PV+ and SCGN-/PV+ neurons presented by the authors and the respective coupling to cortical activity states does show partial overlap and argues rather for gradual differences between striatal interneurons than presenting a strong case for categorical differences – as for comparison is well established for distinct PV+ subpopulations in hippocampus. Also, the differential innervation of putative direct versus indirect pathway medium spiny neurons is another interesting starting point of an in depth study evaluation the functional consequences of this finding. In summary, this study is a – relatively loose – collection of interesting findings – however, none of them is really followed up to reveal important novel insights about distinct functional classes of striatal interneurons – expected to be delivered to warrant publication in a top journal and of interest to wide readership. As it stands, it is a useful data collection for experts interested in novel tools to segregate striatal interneurons.

We agree with the reviewer that the comprehensive classification of cell types often requires one to progress further than the analysis of a small set of molecular markers and thus, to employ a combined analysis of other functional properties such as electrophysiological activity and connectivity. We emphasise that this combined and correlative analysis is exactly what we have provided here for PV+ interneurons of the striatum. The advanced multidisciplinary approach we have used is not only rigorous and quantitative, but is also particularly well suited to deployment in studies of wildtype animals. Indeed, we view the comparative use of wildtype animals of three species to be a key strength of our paper. One can imagine using additional techniques if the appropriate genetically-altered animals were available for study (but, as we highlight in the Discussion, they are not available).

Although single-cell transcriptomics would add a new dimension, such an approach is not necessary or feasible in this study. With regards to necessity, our new hierarchical cluster analysis (see above and Figure 10) has confirmed that not only can the PV+ interneuron population be segregated on the basis of physiological parameters alone, but also that this segregation correlates very well with the selective expression of just one additional marker, that is, secretagogin. Furthermore, our new anatomical analyses (see above and new Figure 2) has verified that the non-uniform spatial distribution of all PV+ interneurons in rat striatum can be explained by selective Scgn expression. Thus, in support of our original conclusions, selective Scgn expression is sufficient for discriminating between functionally diverse subpopulations of PV+ interneurons. With respect to feasibility, it would not be possible to complete a transcriptomics analysis of striatal interneurons of three different species in a timely manner. Again, we are mindful that many applications of single-cell transcriptomics in the central nervous system rely on the use of genetically-altered animals. Indeed, laser-capture microdissection of striatal neurons in double-immunolabeled fixed tissue (as would currently be required to identify PV+/Scgn+ and PV/Scgn- neurons in wildtype animals) would be extremely challenging.

The differential regional distributions of PV+/Scgn+ and PV+/Scgn- has provided some key new insights, especially when one considers the topographical organisation of corticostriatal projections. We have endeavoured to be clear on this point in the Discussion. In further exploring this discovery, we have now tested and verified the hypothesis that the highly unusual distribution of PV+/Scgn+ interneurons can explain the non-uniform distribution all PV+ interneurons in striatum. Thus, our data will give further momentum to a growing notion that not all cell types are distributed equally in the striatum and thus, that the striatum is likely composed of a series of functionally-specialized ‘domains’.

Yes, there is partial overlap in the physiological properties of PV+/Scgn- and PV+/Scgn+ interneurons. We have been clear to highlight this in the paper. Nevertheless our new cluster analysis shows that, based on electrophysiological parameters alone, PV+/Scgn- and PV+/Scgn+ interneurons segregate to the same extent as other interneuron types that experts agree are enormously different (i.e. cholinergic interneurons and NOS+/NPY+ interneurons). It is a challenge to determine whether PV+ interneurons of the hippocampus would yield similarly clear results when subjected to a hierarchical cluster analysis; to our knowledge, we are the first to perform such an analysis on multiple types of interneurons recorded in vivo.

With respect to the functional consequences of a selective innervation of dSPNs and iSPNs, we have now followed up on our original observations by: (1) stating two firm predictions in the context of feedforward inhibition of SPNs; and (2) testing them with new vivo recordings of identified SPNs. See Results subsection “PV+/Scgn- and PV+/Scgn+ interneurons differ in their temporal relationships with striatal projection neurons of the direct and indirect pathways” and new Figure 9. As we have detailed, our analysis of these new SPN recordings verified the prediction for PV+/Scgn- interneurons. The analysis did not validate a second prediction for PV+/Scgn+ neurons, but instead supported an important new concept, namely that PV+/Scgn+ interneurons might not be as well positioned as PV+/Scgn- interneurons to provide classical feedforward inhibition in the striatal microcircuit.

With our extensively revised paper, we have good reason to suggest that it provides the most detailed correlative ex vivo/in vivo analyses of the properties of striatal PV+ interneuron types to date. We have detailed a large number of novel insights. We have thus delivered much evidence of our paper being a cohesive account of important discoveries that will be of great interest to a wide audience.